# Mapping the broadband circular dichroism of copolymer films with supramolecular chirality in time and space

Marius Morgenroth[1], Mirko Scholz[1], Min Ju Cho[2], Dong Hoon Choi[2], Kawon Oum [1✉] & Thomas Lenzer [1✉]

Measurements of the electronic circular dichroism (CD) are highly sensitive to the absolute configuration and conformation of chiral molecules and supramolecular assemblies and have therefore found widespread application in the chemical and biological sciences. Here, we demonstrate an approach to simultaneously follow changes in the CD and absorption response of photoexcited systems over the ultraviolet—visible spectral range with 100 fs time resolution. We apply the concept to chiral polyfluorene copolymer thin films and track their electronic relaxation in detail. The transient CD signal stems from the supramolecular response of the system and provides information regarding the recovery of the electronic ground state. This allows for a quantification of singlet—singlet annihilation and charge-pair formation processes. Spatial mapping of chiral domains on femtosecond time scales with a resolution of 50 μm and diffraction-limited steady-state imaging of the circular dichroism and the circularly polarised luminescence (CPL) of the films is demonstrated.

[1] Department Chemistry and Biology, Physical Chemistry 2, Faculty IV: School of Science and Technology, University of Siegen, Adolf-Reichwein-Str. 2, 57068 Siegen, Germany. [2] Department of Chemistry, Research Institute for Natural Sciences, Korea University, 145 Anam-ro, Seongbuk-gu, Seoul 02841, Republic of Korea. ✉email: oum@chemie.uni-siegen.de; lenzer@chemie.uni-siegen.de

**B**roadband electronic circular dichroism spectroscopy, measuring the difference in optical density (OD) between left-circularly polarised (LCP or shortly L) and right-circularly polarised (RCP or shortly R) light (i.e. $CD = OD_L - OD_R$), is a powerful method to discriminate between molecular systems of different chirality. These include compounds and assemblies which behave as image and mirror-image (enantiomers) and diastereomers bearing several chiral centres. Chemical chirality occurs from very small to very large length scales, ranging from small molecular systems, e.g. naturally occurring D-sugars and L-amino acids, up to supramolecular structures, such as DNA or cholesteric polymer assemblies. As a result, steady-state CD spectroscopy has been widely applied to chemical and biological systems for obtaining absolute structural information[1-5].

Time-resolved CD (TrCD) spectroscopy has a high potential to elucidate changes in chirality on ultrashort time scales[6,7]. Yet, applications employing spectrally broadband CD detection covering the dynamics down to the femtosecond time-scale are still scarce, likely due to the small transient signal of many CD-active systems and nontrivial issues regarding polarisation management. Notable exceptions include the transient response recorded for merocyanine nanorod aggregates over the spectral range $360-550$ nm with subpicosecond time resolution by Fiebig and co-workers[8] and investigations in the UV range ($250-370$ nm) of an enantiopure ruthenium complex and a chiral synthetic thioamide-substituted dipeptide with 500 fs temporal resolution of Oppermann et al.[9,10]. Whereas in the former case, polarisation switching was achieved by a Pockels cell, the latter approach employed a photoelastic modulator. As another method, polarisation mirroring was introduced by Brixner and co-workers. There, an LCP pulse and its RCP mirror image were generated using an arrangement of mirrors and periscopes[11,12]. All of these approaches directly determine the transient CD signal as the difference in OD for probing with LCP and RCP light ($\Delta CD = \Delta\Delta OD = \Delta OD_L - \Delta OD_R$). Alternatively, $\Delta CD$ can be obtained using self-heterodyned detection, where the chiral response is extracted from the free induction decay of the optical activity. This technique was introduced by Cho and co-workers[13-15] and later on applied by Hiramatsu and Nagata[16] to enantiopure $[Ru(bpy)_3]^{2+}$ as a test case, but further experiments of this type have not yet been reported.

Chiral copolymer thin films, such as poly({9,9-bis[(3S)-3,7-dimethyloctyl]fluorenyl-2,7-diyl}-*alt*-{benzo[2,1,3]thiadiazol-4,8-diyl}), shortly c-PFBT (Fig. 1a), are worthwhile targets for studies using TrCD and transient absorption (TA) methods. Such films are of considerable interest because they exhibit an extraordinarily strong CD response in the range of several degrees and also emit strong circularly polarised luminescence[17-19]. This makes them attractive for applications such as colour-tuning devices or polymer-based OLEDs[20-22]. The strong optical activity of these films is typically assigned to a long-range cholesteric (=chiral nematic) ordering[17], yet this interpretation was very recently challenged by Campbell, Fuchter and co-workers, who assigned the origin of the strong chiroptical response of c-PFBT to the coupling of molecular magnetic and electric moments (i.e. single-molecule optical activity)[19,23].

In this work, we present an advanced setup for TrCD detection using a Pockels-cell-based polarisation-switching design employing a deuterated KDP crystal (replacing the previously used BBO[24,25]). Consequently, the time resolution is improved to 100 fs, with the two decisive benefits of a substantial increase in the intensity of the circularly polarised multifilament supercontinuum over the spectral range covered ($260-700$ nm) and an accompanying reduction of its intensity fluctuations. Simultaneous detection of TrCD and TA signals ($\Delta OD = 0.5 \cdot [\Delta OD_L +$ $\Delta OD_R]$) from four consecutive probe laser shots (LCP and RCP, each with and without the pump beam) is demonstrated. We obtain TrCD and steady-state CD spectra for c-PFBT thin films which strongly support a supramolecular origin of the large CD response. In addition, TrCD mapping of such thin-film structures is important for understanding the dynamics in different chiral domains, and we demonstrate here a proof-of-concept implementation showing ultrafast CD mapping on the femtosecond time scale with a spatial resolution of 50 μm, as well as steady-state CD and CPL imaging with a diffraction-limited resolution of about 500 nm.

## Results and discussion

**Steady-state absorption, CD and CPL spectroscopy.** The c-PFBT copolymer was deposited on borosilicate glass slides without an alignment layer and thermally annealed at 150 °C, resulting in yellow films (see photographs in Fig. 1a). Such films have multidomain cholesteric liquid crystalline order with the statistical orientation of the individual domains[20,26]. Steady-state absorption spectra recorded with LCP and RCP light showed substantial differences (blue and red lines in the top panel of Fig. 1b), resulting in a strong CD response with OD values of up to $-0.3$ ($=-10,000$ mdeg) and a dissymmetry factor $g_{abs} = 2(OD_L - OD_R)/(OD_L + OD_R)$ approaching $-0.5$ (middle and bottom panels of Fig. 1b). In addition, the films displayed strongly polarised photoluminescence with a dissymmetry factor $g_{lum} = 2(I_L - I_R)/(I_L + I_R)$ of up to $-0.4$ (Fig. 1b, brown lines).

Importantly, the thin films showed virtually no changes in the CD response and the dissymmetry factor $g_{abs}$ upon turning or flipping of the sample (Fig. 1c). Therefore, we conclude that there are no significant contributions resulting from a combination of linear anisotropies (i.e. linear dichroism and linear birefringence) in our sample in combination with any possible anisotropies in the spectroscopic setup. We also observed that the CD and $g_{abs}$ values of c-PFBT increased considerably with film thickness, in agreement with previous observations of Abbel et al.[17]. This is demonstrated in Fig. 1d for three films with thicknesses $d$ of $62 \pm 5$, $106 \pm 8$ and $241 \pm 16$ nm (blue, red, and black lines, respectively).

**Contactless laser-based determination of film thickness.** The film thickness was determined in an all-optical, non-invasive fashion using laser-based picosecond ultrasonics[27-29] from the TA traces of our ultrafast measurements. The TA kinetics of the polymer films exhibited pronounced damped oscillations at the edges of the ground-state absorption bands (see Fig. 1e) (pump wavelength 320 nm, probe wavelength range $470-510$ nm). The oscillation in the absorption signal arises from a coherent acoustic phonon, which is induced by the pump laser pulse and propagates back and forth between the polymer−glass and polymer−nitrogen interface. The period of this oscillation is directly proportional to the thickness of the film ($d = \tau_a \cdot c_L/4$), where $\tau_a$ is the measured oscillation period of ($100 \pm 2$), ($171 \pm 3$) and ($388 \pm 3$) ps for the films investigated, and $c_L$ is the longitudinal sound velocity of c-PFBT, which we previously determined as ($2490 \pm 150$) m s$^{-1}$[29]. The inset of Fig. 1e shows that the $g_{abs}$ value is very small for the thinnest film and then strongly increases with the thickness ($g_{abs}$ peak values of 0.028, 0.215, and 0.476, respectively). Such behaviour suggests a supramolecular (nonlocal) origin of the CD response in this thickness range[17,20,26]. In contrast, single-molecule circular dichroism, where coupling between local electric and magnetic transition dipole moments is operative, should lead to a thickness-independent $g_{abs}$[17,19].

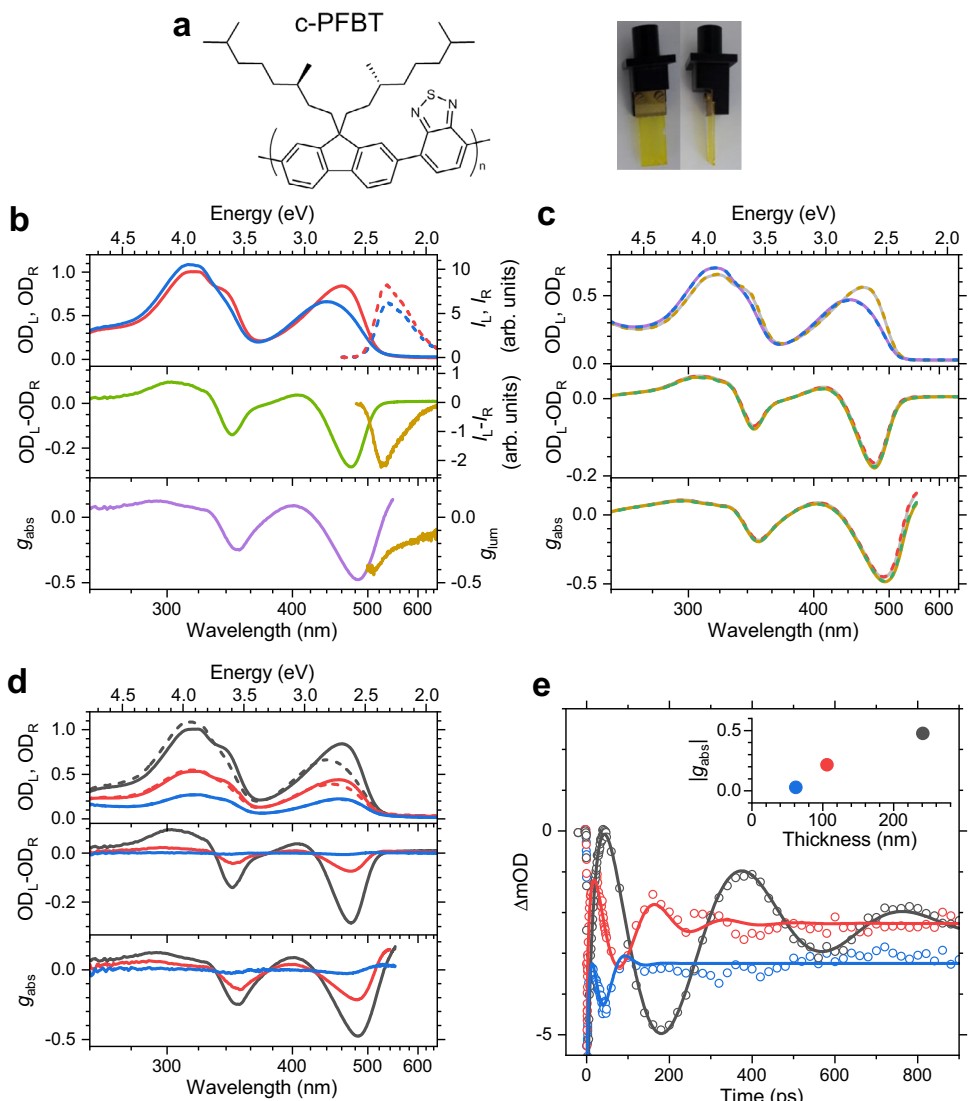

**Fig. 1 Optical properties of the c-PFBT thin films. a** Chemical structure of the c-PFBT copolymer and pictures of the thin film on glass. **b** Absorption spectra for left-circularly (blue solid line) and right-circularly polarised light (red solid line); emission spectra for left-circularly (blue dashed line) and right-circularly polarised light (red dashed line); CD spectrum (green line) and CPL spectrum (brown line); dissymmetry parameters $g_{abs}$ (violet line) and $g_{lum}$ (brown line). **c** Behaviour upon flipping and turning of the film: Absorption spectra for left-circularly polarised light (front side: grey solid line, backside: brown dashed line) and right-circularly polarised light (front side: violet solid line, backside: blue dashed line); CD spectra and $g_{abs}$ (front side: grey solid line, front side turned by 180°: red dashed line, backside: brown solid line, backside turned by 180°: green dashed line). **d** Thickness-dependent absorbance (left-circularly polarised light: dashed lines, right-circularly polarised light: solid lines), as well as CD and $g_{abs}$ for thin films with the thickness 62 nm (blue), 106 nm (red) and 241 nm (black). **e** Transient absorption signals averaged over the probe wavelength range 470−510 μm (open circles) showing oscillations due to a coherent acoustic phonon excited by the 320 nm pump laser pulse for thin films with the thickness 62 nm (blue), 106 nm (red) and 241 nm (black); solid lines represent the respective fits using a sum of a biexponential function and a damped cosine function; the inset shows the absolute peak value of $g_{abs}$ as a function of film thickness, suggesting a supramolecular origin of the chiroptical response.

**CD and CPL imaging with diffraction-limited resolution.** To further characterise the chiroptical properties of the c-PFBT films, a microscopy setup for CD and CPL imaging was constructed. Several powerful setups for steady-state CD imaging have been established so far, starting from the early work of Maestre and Katz[30]. These include circular dichroism imaging based on a wide-field microscope featuring illumination through a combination of an interference filter, a polariser and a tunable quarter-wave retarder[31], scanning CD optical microscopy using a polariser and a photoelastic modulator[32], CD microscopy with discretely modulated circular polarisation[33], as well as scanning UV–Vis circular dichroism experiments employing highly collimated synchrotron radiation[34,35]. Our specific implementation

based on wide-field microscopy reaches a spatial resolution of ca. 500 nm and has the additional option to simultaneously record CD and CPL spectra integrated over the entire field of view. Representative results for different areas ($80 \times 60\,\mu m^2$) of a c-PFBT film (thickness ca. 240 nm) are displayed in Fig. 2.

The CD image in panel a and the corresponding $g_{abs}$ image in panel b (175,000 pixels each) was obtained at the peak of the CD spectrum using a 10 nm narrow bandpass filter with a centre wavelength of 470 nm. The graphs in panel c show the spectrally resolved data integrated over the complete field of view (OD for LCP and RCP light, CD and $g_{abs}$ spectrum). The film area shows island-like structures and has a granular appearance, in agreement with microscope images obtained for a c-PFBT film between

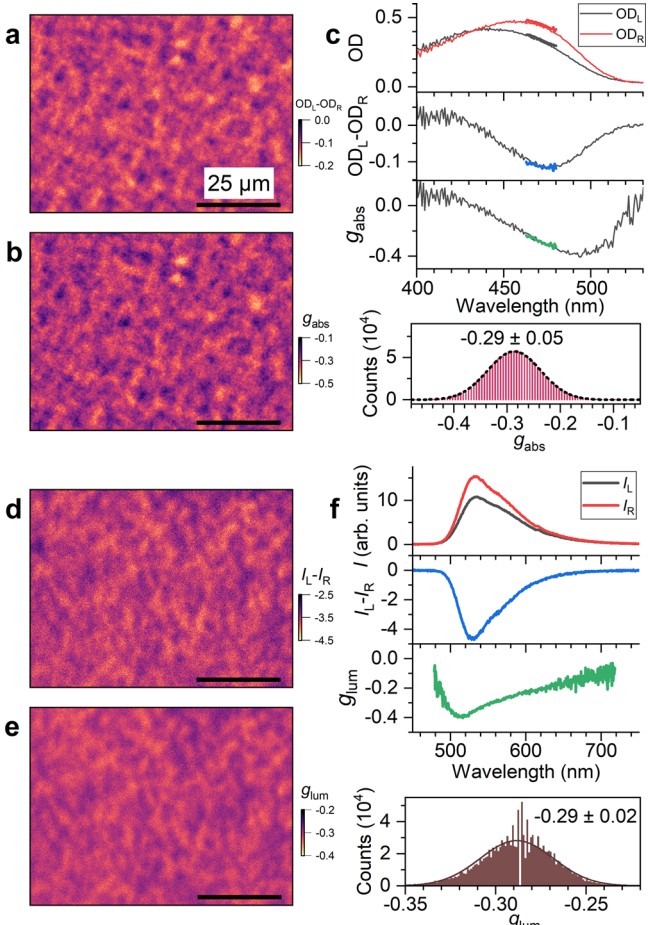

**Fig. 2 CD and CPL imaging with diffraction-limited resolution of about 500 nm. a** Microscope image ($80 \times 60 \, \mu m^2$) for the circular dichroism of a c-PFBT thin film. **b** Corresponding image for the dissymmetry factor $g_{abs}$. **c** Wavelength-dependent optical density for LCP (black) and RCP detection (red), the resulting CD signal and the $g_{abs}$ spectrum (thin lines: full spectra, thick coloured lines: spectral region selected by the bandpass filter at 470 nm used for CD imaging) as well as the $g_{abs}$ histogram with a Gaussian fit (dashed black line), all determined over the entire field of view ($210 \times 160 \, \mu m^2$). **d** Microscope image for the circularly polarised luminescence at another position on the same c-PFBT thin film. Excitation at 365 nm. **e** Corresponding image for the dissymmetry factor $g_{lum}$. **f** Wavelength-dependent photoluminescence spectrum for LCP (black) and RCP detection (red), the resulting CPL signal and the $g_{lum}$ values for the c-PFBT emission band, as well as the $g_{lum}$ histogram with a Gaussian fit (solid black line), all determined over the entire field of view. The length of the black scale bar in each image corresponds to 25 μm.

crossed polarisers from a previous study of Abbel et al.[17], indicating a disordered multidomain liquid crystalline arrangement. The underlying structure was assigned recently to a long-range hierarchical arrangement of fibrils[36]. The statistics over the entire field of view provides a histogram (bottom right), which is well described by a Gaussian distribution with $g_{abs} = -0.29 \pm 0.05$. For further illustration of the capabilities of diffraction-limited CD imaging, we provide additional images for other regions of the same film in Supplementary Note 1. We also checked the invariance of the CD images upon sample rotation and flipping, as illustrated in Supplementary Note 2, which proved that there are also no significant contributions on micrometre length scales resulting from a combination of linear dichroism and linear birefringence in our sample in combination with any possible

anisotropies of the CD microscopy setup. In addition, the structures revealed in the CD images are very similar to the structures observed using a conventional crossed-polariser arrangement (Supplementary Note 3).

Panels d and e contain the CPL and $g_{lum}$ images for a different position on the same film obtained upon photoexcitation at 365 nm. The graphs in panel f show the spectrally resolved data integrated over the complete field of view (PL for LCP and RCP light, CPL and the $g_{lum}$ spectrum, all in good agreement with the data shown in Fig. 1b). The appearance of the CPL and $g_{lum}$ images is very similar to those obtained from CD imaging, i.e. they show the same island-type areas. Regions with larger absolute values of $g_{lum}$ also show larger absolute values of $g_{abs}$. The statistics on the $g_{lum}$ image provides a histogram, which can be again well described by a Gaussian distribution with $g_{lum} = -0.29 \pm 0.02$. The size of the structures seen both in the $g_{abs}$ and $g_{lum}$ images correlate well with the dimensions of fibre-type and aggregated spherulite arrangements reported by Di Nuzzo et al. and Lakhwani and co-workers for disordered multidomain films of c-PFBT[20,36].

To further highlight the capabilities of the method, we present in Fig. 3a–d CD imaging results for different areas ($80 \times 60 \, \mu m^2$) of another c-PFBT thin film. In this example, we intentionally selected four regions across the film (also in less homogeneous regions) where the CD images show pronounced differences. The shape of the CD spectra integrated over all regions is quite similar, with maximum OD amplitudes between $-0.21$ ($-7000$ mdeg) and $-0.30$ ($-10,000$ mdeg), as shown on the right side. The corresponding $g_{abs}$ images are presented in panels e–h. The similar colours in panels e and h compared with the more disparate colours in a and d show that the stronger CD in panel a is due to the larger OD. Normalisation to the OD provides similar $g_{abs}$ values in both cases, as shown in the histograms on the right side ($-0.37 \pm 0.02$ vs. $-0.36 \pm 0.03$). Still, the island-like structures observed in panel a are more extended than those in panel d, suggesting slight differences in the structure of both film regions.

Next, we compare panels a and b. The peak value of the CD spectrum in b is about 5% larger ($-0.30$ vs. $-0.29$ in OD). However, in this case, the difference in $g_{abs}$ is 20% ($-0.44 \pm 0.03$ vs. $-0.37 \pm 0.02$), suggesting that the larger OD is not the main reason for the stronger circular dichroism in b. As the last pair, we compare panels c and d. The peak value of the CD spectrum in d is about 17% smaller ($-0.21$ vs. $-0.25$ in OD), yet $g_{abs}$ are only smaller by about 8% ($-0.36 \pm 0.03$ vs. $-0.39 \pm 0.05$) showing that the difference in $g_{abs}$ is only partly due to the different OD of the two regions. The most amazing difference is however the totally different appearance of the image in panel g. Here we find a mosaic structure, where micrometre-size regions with large absolute $g_{abs}$ (yellow) are very close to micrometre-size regions with much smaller $g_{abs}$ (violet). The distribution in panel g also has the largest width ($-0.39 \pm 0.05$) and is skewed, with pronounced deviations from Gaussian behaviour. These four examples show the potential of diffraction-limited CD imaging to address differences in chiroptical properties with a high spatial resolution for systems, which macroscopically show quite similar averaged CD spectra. It could well be that the different appearance in less homogeneous regions points toward different aggregation pathways, as previously suggested for oligothiophene films[34,35]. Still, the thin film regions in the centre of the substrate and at intermediate distances are very uniform and also microscopically homogeneous, with no clear indications for different aggregation pathways.

**Ultrafast transient CD and TA experiments.** Figure 4a displays typical results of the combined ultrafast TrCD/TA experiment, in

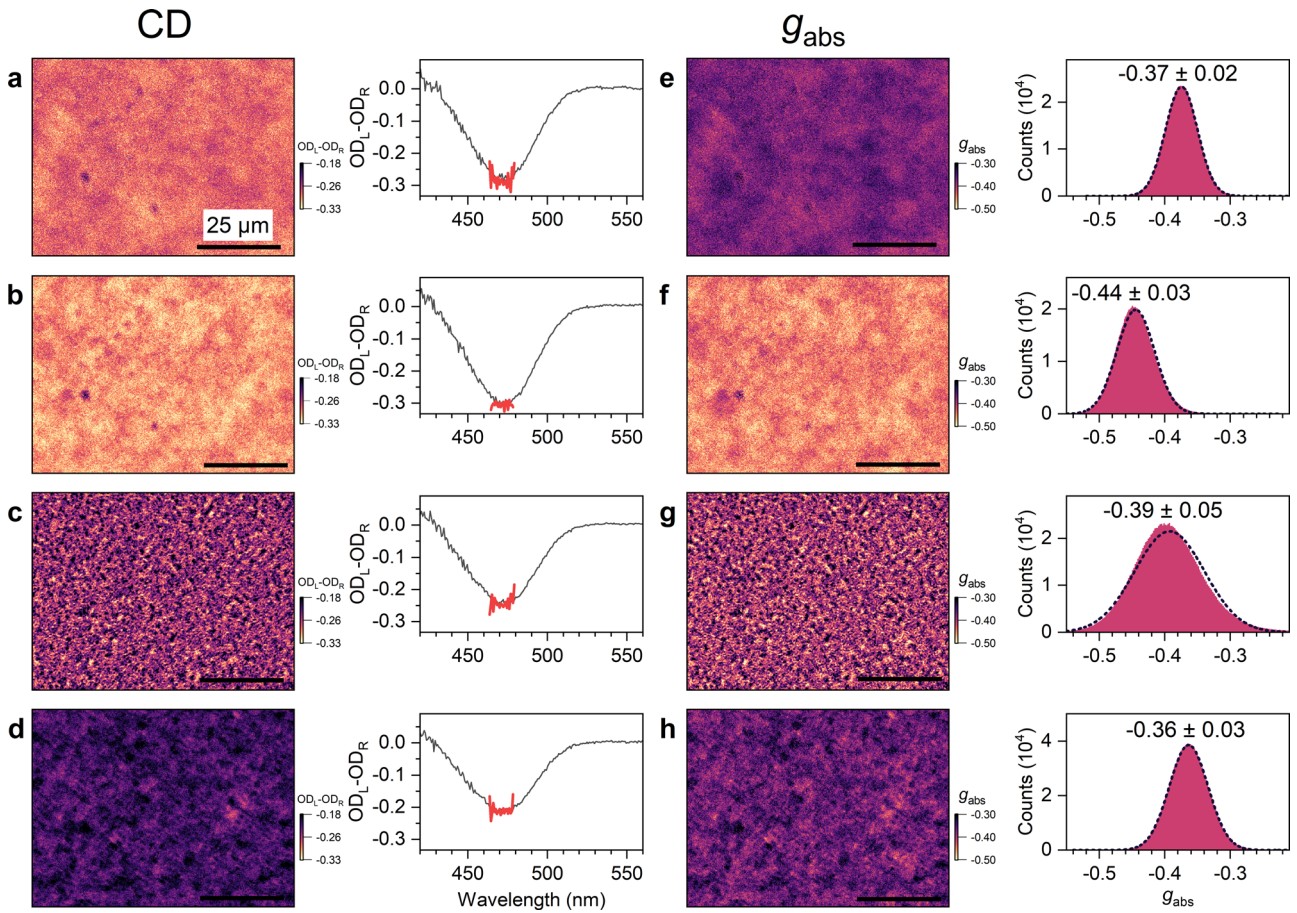

**Fig. 3 CD imaging with diffraction-limited resolution for different regions of another c-PFBT thin film. a–d** Microscope images (80 × 60 μm²) for the circular dichroism of the c-PFBT thin film at different positions. Plots on the right show the CD signals integrated over the entire field of view (210 × 160 μm², thin black lines: full spectra, thick red lines: spectral region selected by the bandpass filter at 470 nm used for CD imaging). **e–h** Corresponding images for the dissymmetry factor $g_{abs}$, with histograms on the right including Gaussian fits as dashed black lines, determined over the entire field of view. The length of the black scale bar in each image indicates a distance of 25 μm. Variations in counts in the $g_{abs}$ distributions are due to the different bin sizes used.

which a 241 nm thin c-PFBT film was excited by 50 fs pump laser pulses at 320 nm. Here we focus on the TA and TrCD spectra at six representative time delays (see Supplementary Note 4 for contour plots of the complete data sets). TA spectra for probing with LCP and RCP light are shown as blue and red lines. At early times (0.2 ps in Fig. 4a), the TA spectra show pronounced ground state bleach (GSB) features centred at about 450 and 315 nm, as the pump beam initially promotes ground state ($S_0$) molecules to a higher excited singlet state $S_x$. These bands resemble the corresponding inverted steady-state absorption spectra for LCP and RCP probing (cf. Fig. 1b, top). Prominent excited state absorption (ESA) bands emerge at 620, 530 (shoulder), 365 and 284 nm. The differences in the TA spectra for LCP and RCP probing give rise to a pronounced TrCD difference signal, which is indicated by the green line. This giant TrCD signal closely resembles the inverted steady-state CD spectrum (Fig. 1b, middle). Remarkably, it does not show any clear sign of CD activity in the excited electronic state, see e.g. the region 360−380 nm, where the virtually silent TrCD signal is in marked contrast to the substantial ESA band of the TA spectrum. We take this important observation as a strong indication of the supramolecular nature of the CD response, as described in more detail below. The flat and weak, slightly negative TrCD signal above 520 nm, where c-PFBT does not have any ground-state absorption, is largely due to circular selective

scattering[5,20,37,38], and a similar weak feature is also observed in the ground-state CD spectrum.

The remaining panels in Fig. 4a show that at later times (0.5−980 ps) all bands in the TA and TrCD spectra decay. While the TrCD spectra exhibit essentially no changes in shape, the TA spectra for LCP and RCP probing show distinct differences. In the spectral range above 500 nm, the peak/shoulder structure at early times evolves into a flatter ESA band at later times (see the spectra at 100 and 980 ps). As will be discussed below in the kinetic analysis, the flat ESA band can be explained by the formation of a charge-pair state at early times.

Figure 4b contains spectra for different pump fluences, with initial exciton number densities $N_0(S_x)$ in the range $5.7 \times 10^{17}$–$1.2 \times 10^{19}$ cm$^{-3}$ at the fixed pump−probe delay time of 3 ps. In the two upper panels, the amplitude of the TA signals for probing with left-circularly and RCP light increases with increasing fluence below 500 nm. This is accompanied by a shift of the main GSB band from 455 to 445 nm. In contrast, above 500 nm there is a change in sign: At low pump fluence, there is a negative band below 580 nm, which we assign to stimulated emission from $S_1$, whereas at high pump fluence there is absorption in the entire 500−650 nm range. As explained in more detail below, the latter effect is again consistent with the formation of a charge-pair state from $S_1$ by singlet−singlet

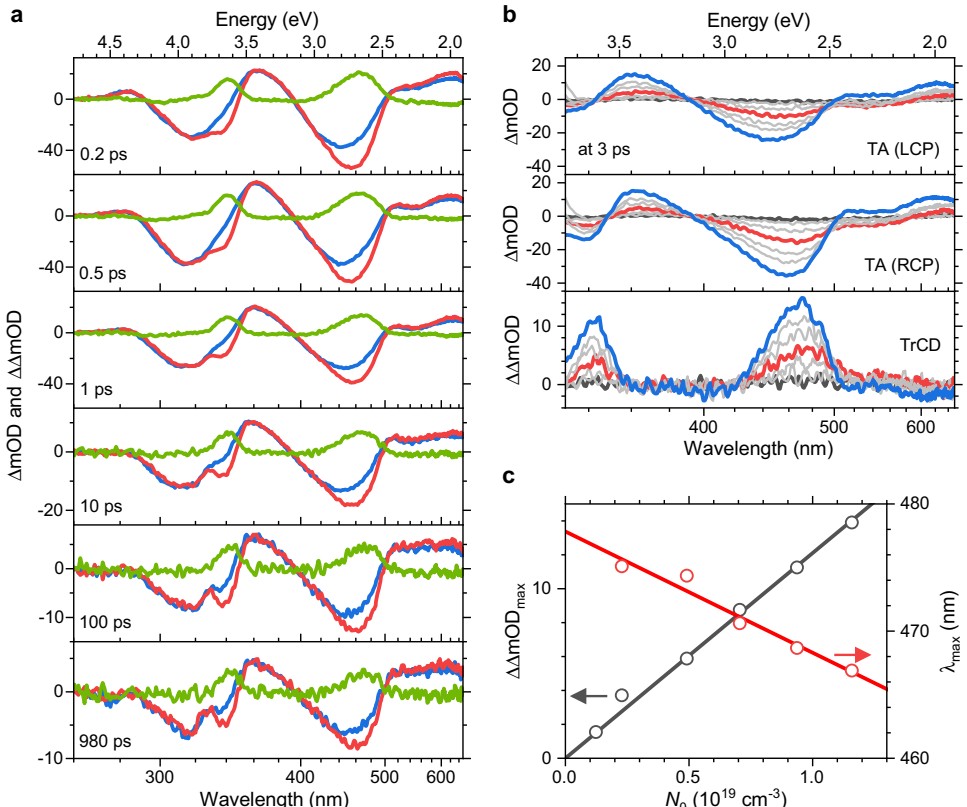

**Fig. 4 Ultrafast transient circular dichroism and transient absorption spectra of a 241 nm thin c-PFBT thin film. a** Spectra at six different delay times: transient absorption for probing with left-circularly polarised light (blue), right-circularly polarised light (red) and the resulting TrCD spectrum (green). **b** Fluence-dependent spectra at 3 ps for initial exciton number densities between $5.7 \times 10^{17}$ (black) and $1.2 \times 10^{19}$ cm$^{-3}$ (blue); red line for $N_0 = 4.9 \times 10^{18}$ cm$^{-3}$; left-circularly polarised probe light (top), right-circularly polarised probe light (middle) and the resulting TrCD spectrum (bottom). **c** Correlation of the TrCD peak signal (black circles) and the peak position of the TrCD signal (red circles) with the initial exciton number density including a proportional (black) and a linear (red) fit.

annihilation (SSA). This process becomes faster at high pump fluence, and thus high initial exciton number densities in the polymer[25,39], as described by our kinetic modelling below. Therefore, under such high-fluence conditions, the $S_1$ population is already considerably depleted at 3 ps.

We are now coming to the corresponding fluence-dependent TrCD spectra, which are depicted in the bottom panel of Fig. 4b. The TrCD signal amplitude depends linearly on the pump fluence (Fig. 4c, black open circles and fit line). Interestingly, the main band of the TrCD spectrum at 475 nm also shows a fluence-dependent blue shift as observed for the TA spectra. This shift in wavelength is illustrated by the red open circles and the corresponding red fit line in Fig. 4c. We explain this as follows: At low pump fluence, the $S_1$ state gives rise to circularly polarised stimulated emission (CPSE), i.e. a difference in the induced $S_1$ emission for LCP and RCP probing. This process can be viewed as the analogue of CPL, which is however a spontaneous process. CPSE (as CPL) must have the same sign as the corresponding lowest-energy CD band, and as this transient CD bleach of c-PFBT has a positive sign, the CPSE response must also be positive. Indeed, such a CPSE feature is observed, e.g. in the red-coloured TrCD spectrum above 500 nm. This overlapping CPSE band will therefore lead to an apparent broadening of the band toward higher wavelengths at low pump fluence. At the same time, it compensates for the negative contribution above 500 nm, which is the only contribution in this spectral range at high pump fluence (cf. the blue TrCD spectrum above 500 nm in the bottom panel of Fig. 4b) and is due to circular selective scattering. At high pump fluence, CPSE at 3 ps will be largely absent because of fast

depletion of the $S_1$ state due to singlet−singlet annihilation, leading to an apparent blue-shift of the band in the TrCD spectrum[25]. Note that the noise characteristics of the TA and TrCD signals (and thus also the CPSE contribution) are governed by laser noise, i.e. shot-to-shot fluctuations of the intensity and spatial profile of the laser pulses[40]. In contrast, measurements of spontaneous CPL are limited by Poisson noise.

One central question is, why a clear CD bleach signal, as well as CPSE from the $S_1$ state, is observed, whereas one does not see any clear sign of excited-state CD bands in the TrCD spectra. We take this particular finding as a strong indication for the supramolecular origin of the TrCD response, as outlined in the following. Figure 5a shows an idealised sketch of a cholesteric helical arrangement consisting of five layers, each containing 25 c-PFBT units indicated as cylinders, with a side view of the planes on the left, and a view along the twist axis on the right. In the TrCD experiment, between 3% and 15% of the c-PFBT units are initially photoexcited, as can be easily estimated from the ratio of the initial bleach amplitude of the TA spectra and the steady-state absorbance of the sample.

In Fig. 5a we, therefore, choose an excitation level of 10% for an illustration, which is indicated by a statistical selection of 13 excited c-PFBT units in the five planes (excited: red, not excited: blue). One can well understand that photoexcitation makes about 10% of the $S_0$ chromophores inaccessible for the probe beam, which leads to the observed CD bleach feature in the TrCD spectra. In addition, the different electronic properties of the excited species will somewhat modify the in-plane and across-planes longer-range coupling between the electric and magnetic

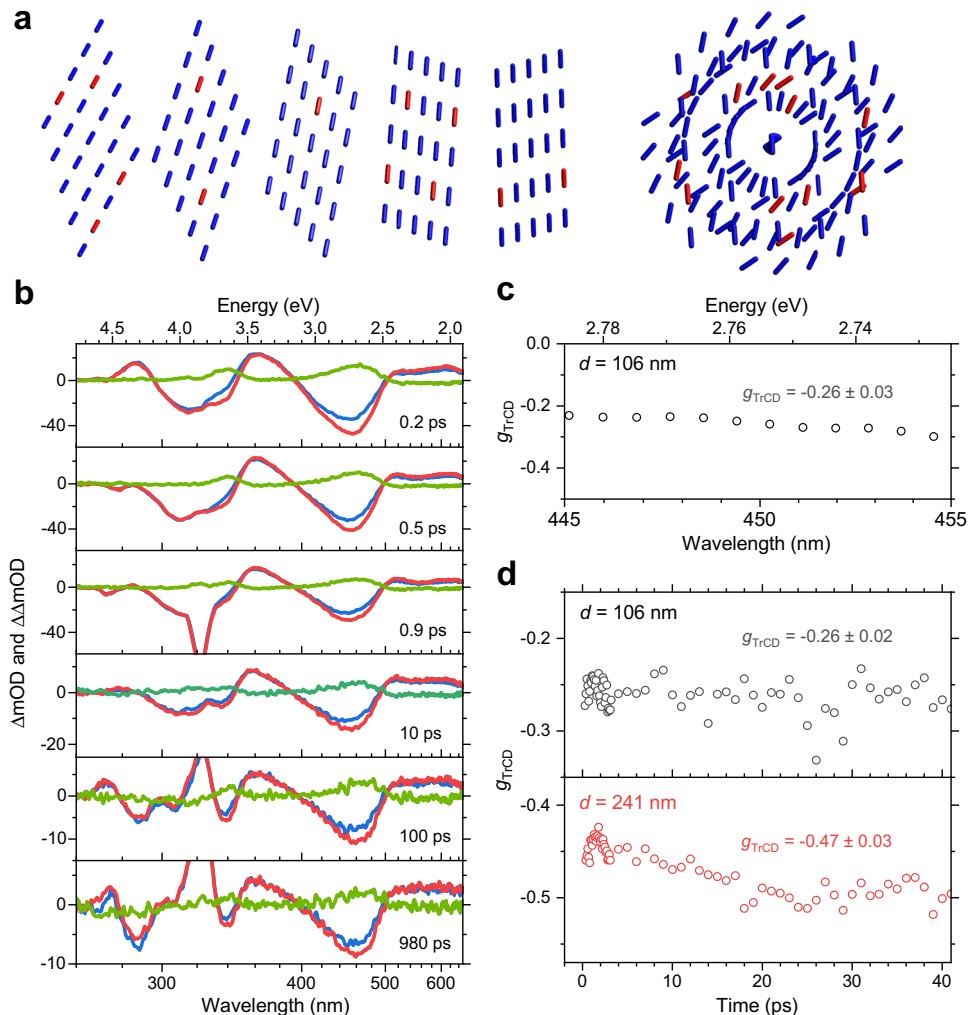

**Fig. 5 Supramolecular origin of the ultrafast TrCD response of c-PFBT thin films. a** Schematic representation of a cholesteric helical arrangement consisting of five layers with 25 c-PFBT units, where ground state c-PFBT units are shown as blue cylinders and excited state c-PFBT units are represented by red cylinders; left side: side view, right side: view along the twist axis. **b** Transient spectra for a 106 nm thin c-PFBT film at six different delay times: transient absorption for probing with left-circularly polarised light (blue), right-circularly polarised light (red) and the resulting TrCD spectrum (green). Note the about 50% smaller TrCD response compared with the 241 nm thin film (Fig. 4a) despite the comparable amplitude of the transient absorption signals. **c** Wavelength-dependence of the dissymmetry factor $g_{TrCD}$. **d** Time-dependence of the dissymmetry factor $g_{TrCD}$ for a 106 nm thin film (top) and a 241 nm thin film (bottom).

transition dipole moments of c-PFBT in the $S_0$ state. However, this effect appears to be weak, as there is a close resemblance of the TrCD spectrum to the inverted steady-state CD spectrum. As already mentioned, CPL is also thought to be a long-range supramolecular effect, likely resulting from processes such as circular intensity differential scattering[20,36] and linearly polarised luminescence of quasi-nematic layers with subsequent circular polarisation by the remaining film layer[41]. Therefore, the CPSE contribution in the TrCD spectra, i.e. directed $S_1$ emission induced by the probe beam of the TrCD experiment, should be based on similar effects as the CPL signal. In contrast, considering CD activity in the excited state, Fig. 5a suggests that efficient long-range coupling between the spatially widely distributed electronically excited c-PFBT units must be weak, both in-plane and across planes.

Further support for the supramolecular origin of the TrCD signal is obtained from thickness-dependent TrCD measurements. Figure 5b contains the results for the 106 nm thin c-PBFT film at the same time delays as in Fig. 4a for the 241 nm thin film. Qualitatively, the TA and TrCD signals look the same for both

films, however importantly, the TrCD signal in Fig. 5b is by about a factor two smaller, although the initial TA bleach amplitude is almost the same as in Fig. 4a. To quantify this effect, we introduce the time-dependent dissymmetry factor $g_{TrCD}$. For that, we normalise the TrCD signal (i.e. $\Delta OD_L - \Delta OD_R$) to the TA signal for unpolarised light (which is $0.5 \cdot (\Delta OD_L + \Delta OD_R)$) resulting in $g_{TrCD} = 2(\Delta OD_L - \Delta OD_R)/(\Delta OD_L + \Delta OD_R)$. Figure 5c shows the wavelength-dependent $g_{TrCD}$ averaged over the time interval $0.2-48$ ps in the region of the main peak of the TrCD signal (445–455 nm). We obtain $g_{TrCD} = -0.26 \pm 0.03$ in good agreement with the steady-state value for $g_{abs}$ in Fig. 1d (red line). The time dependence of $g_{TrCD}$ is analysed in Fig. 5d. Interestingly, $g_{TrCD}$ is largely time-independent, where the larger scatter at longer times is simply due to the substantially decreased signal amplitude of the TrCD signal. Corresponding measurements for the 241 nm thin film provide the value $g_{TrCD} = -0.47 \pm 0.03$, again in good agreement with the steady-state value $g_{abs}$ (Fig. 1d, black line). The thickness-dependent $g_{abs}$ and $g_{TrCD}$ values, as well as the absence of CD activity in the excited state, provide strong arguments against a mechanism based on single-molecule

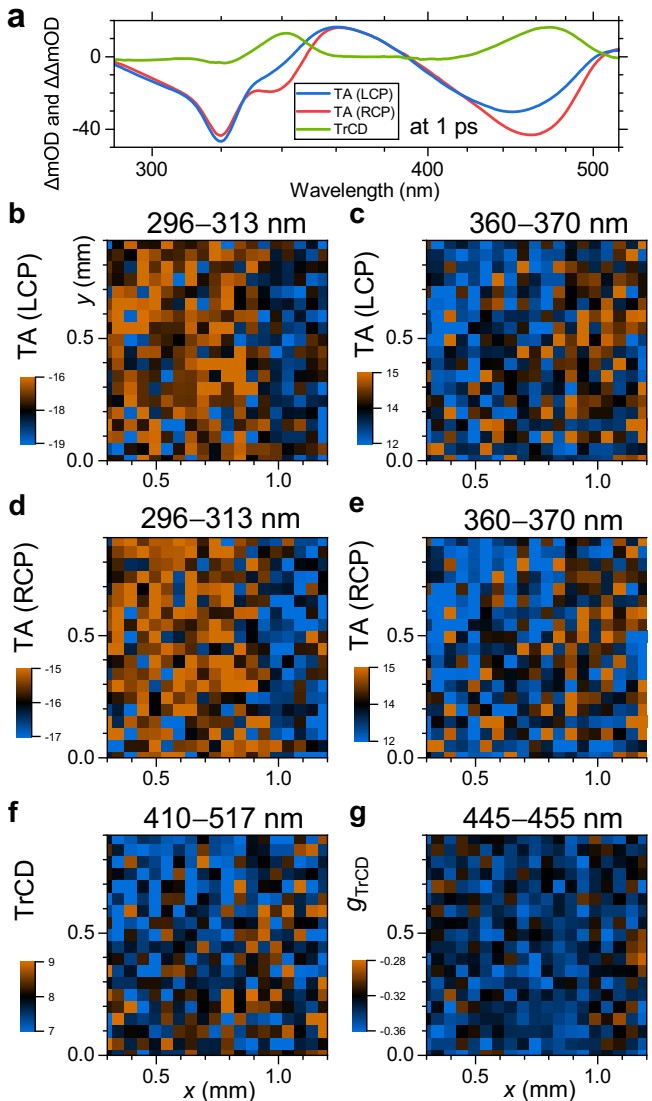

**Fig. 6 Ultrafast transient absorption and circular dichroism mapping of a c-PFBT thin film at 1 ps. a** Transient absorption spectra for probing with left-circularly polarised light (blue) and right-circularly polarised light (red), as well as the resulting TrCD spectrum (green), averaged over the complete map. **b** Transient absorption map for probing with left-circularly polarised light averaged over the wavelength range 296−313 nm. **c** Transient absorption map for probing with left-circularly polarised light (average: 360−370 nm). **d** Transient absorption map for probing with right-circularly polarised light (average: 296−313 nm). **e** Transient absorption map for probing with right-circularly polarised light (average: 360−370 nm). **f** Transient circular dichroism map (average: 410−517 nm). **g** Map for the dissymmetry factor $g_{TrCD}$ (average: 445−455 nm).

circular dichroism, with coupling between local electric and magnetic transition dipole moments, which would predict thickness-independent gabs, $g_{lum}$[19] and also $g_{TrCD}$ as well as pronounced CD activity in the electronically excited state. The TrCD signal for a given film thickness is still linearly dependent on the pump fluence, i.e. it scales with the number of initially excited polymer units (cf. Fig. 4b, c). The supramolecular nature of the transient chiral response manifests itself in terms of an additional constant scaling factor which increases the TrCD signal, which means it becomes much bigger for thicker films.

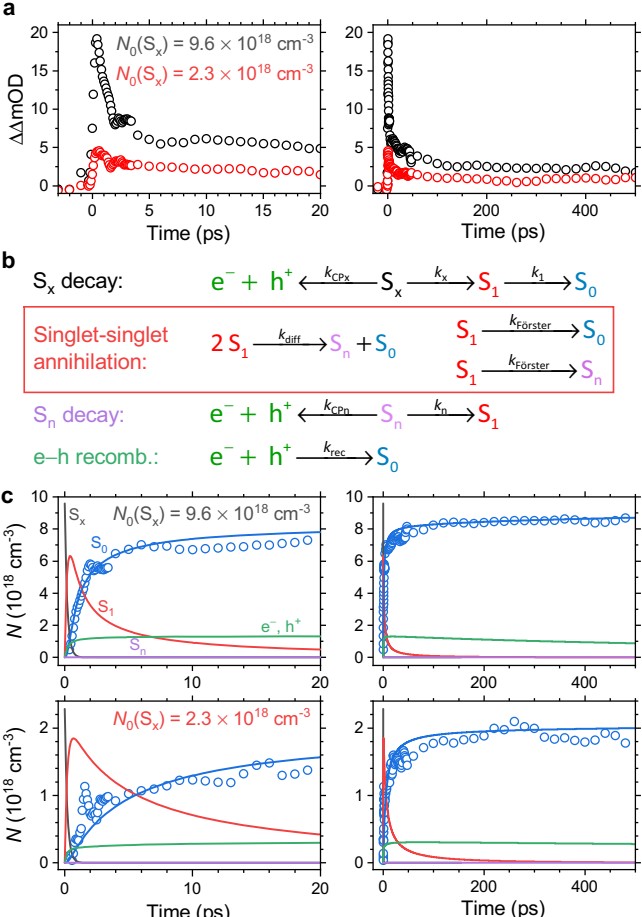

**Fig. 7 Kinetic modelling of the TrCD kinetics of a 241 nm thin c-PFBT film. a** TrCD decay kinetics (averaged over the TrCD peak region 460−470 nm) for the initial $S_x$ exciton number densities $N_0(S_x)$ = $9.6 \times 10^{18}$ cm$^{-3}$ (black) and $2.3 \times 10^{18}$ cm$^{-3}$ (red). **b** Kinetic model for describing the population dynamics. **c** Results of the kinetic modelling for the initial $S_x$ exciton number densities $9.6 \times 10^{18}$ cm$^{-3}$ (top) and $2.3 \times 10^{18}$ cm$^{-3}$ (bottom), with the dynamics on short time scales (up to 20 ps) on the left side and long time scales (up to 500 ps) on the right side. Blue circles: Recovery of $S_0$ population as experimentally determined in the TrCD measurements with corresponding blue fit lines obtained from the kinetic model; black, red, green and violet lines: number density of the $S_x$ excitons, $S_1$ excitons, electron−hole pairs (e$^-$, h$^+$) and the $S_n$ excitons, as obtained from the kinetic model.

**Transient CD and transient absorption mapping.** The high signal-to-noise ratio of the transient TA and TrCD signals also allows for a time-resolved point-by-point mapping of the c-PFBT thin film. The spatial resolution is defined by the probe beam diameter, which is currently 50 μm. Figure 6 shows an example of such a spatial mapping for an area of about 1 mm$^2$ area (roughly 400 points) at the time delay $t$ = 1 ps. Panel a contains TA spectra for right- and left-circularly polarised probing and the resulting TrCD spectrum averaged over the complete area. Transient absorption maps for LCP and RCP probing are presented in panels b−e. These were averaged over two different wavelength intervals: 296−313 nm (GSB, panels b and d) and 360−370 nm (ESA, panels c and e). In both cases, blue colours correspond to smaller ΔOD values and brown colours to larger ones. All the maps for LCP and RCP probing agree very well. Note that the GSB and ESA maps appear inverted because of the opposite sign

of the signals in these wavelength ranges (cf. panel a). Variations across the maps are quite small, on the order of ±10%.

The resulting TrCD map for the range 410−517 nm is depicted in panel f. It shows clear similarities to the LCP and RCP TA maps in panels c and e. This suggests that, for a given film thickness, there is a correlation between the amplitude of the TA and TrCD signals (larger values on the right side and smaller values on the left side of the map). When normalising the TrCD to the TA signal, i.e. plotting the map for the integrated (445−455 nm) dissymmetry parameter $g_{TrCD}$ in panel g, the pattern largely disappears and one observes a more homogeneous distribution, with an average value of $g_{TrCD} = -0.33$. Time-resolved mapping with higher spatial resolution approaching the diffraction limit is planned for future studies.

**Kinetic modelling of TrCD signals.** We finally deal with the quantitative kinetic analysis of the ultrafast TrCD signals. Figure 7a shows the TrCD kinetics (averaged over the TrCD peak region 460−470 nm) after excitation at 320 nm for the two initial exciton number densities $N_0(S_x) = 9.6 \times 10^{18}$ cm$^{-3}$ (black) and $2.3 \times 10^{18}$ cm$^{-1}$ (red), respectively. It is evident that the dynamics do not follow a single exponential decay, and that it becomes much faster at larger values of $N_0(S_x)$, with a pronounced ultrafast decay component. This indicates that there must be higher-order processes besides simple intramolecular $S_1$ decay. The transients also show a long-lived component, which does not decay up the maximum delay time of 1500 ps covered in the current experiments. Very importantly, the TrCD kinetics of c-PFBT below 520 nm are only sensitive to the $S_0$ ground state population and do not show contributions from excited-state species. Thus, the two TrCD signals exclusively reflect the recovery of the $S_0$ population, in contrast to transient absorption experiments, where the signal typically arises from a combination of contributions from several species with different absorption coefficients. Also, the TrCD signal is linearly dependent on the $S_0$ concentration for a given film thickness (cf. Fig. 4b, c). The supramolecular nature of the TrCD response becomes evident for different film thicknesses, where thicker films show much larger TrCD signals, which however still increase linearly with the $S_0$ concentration.

To describe the TrCD kinetics, we employed the kinetic model shown in Fig. 7b, which is further supported by fluence-dependent transient absorption experiments at the pump wavelengths 320 and 450 nm (see Supplementary Note 5). Photoexcitation at 320 nm populates a higher excited singlet exciton state denoted as $S_x$, which decays both by internal conversion to $S_1$ (rate constant $k_x = \tau_x^{-1}$) and by the formation of a long-lived charge-pair state (CP, i.e. electron and hole ($e^- + h^+$), rate constant $k_{CPx} = \tau_{CPx}^{-1}$). The total lifetime of the $S_x$ state $\tau_{x,total} = (k_x + k_{CPx})^{-1}$ was determined as 206 fs from ultrafast transient absorption experiments (see Supplementary Note 5), which also clearly showed the long-lived absorption of the CP state. The $S_1$ exciton state decays to $S_0$. For the lifetime of the $S_1$ excitons ($\tau_1 = k_1^{-1}$) a value of 235 ps was determined from separate transient fluorescence measurements (Supplementary Note 6).

Next, we address possible channels which are responsible for the fast decay of the kinetics in the subpicosecond to picosecond range. These are the two singlet−singlet annihilations (SSA) processes shown in the red box in Fig. 7b. One is the diffusive encounter of two $S_1$ excitons producing a higher excited $S_n$ exciton and an $S_0$ ground state species with the rate constant $k_{diff}$. The other channel is Förster resonance energy transfer (FRET)[42,43], for which the decay of an $S_1$ exciton to $S_0$ simultaneously leads to excitation of a nearby $S_1$ exciton to a higher excited $S_n$ state via a nonradiative dipole−dipole coupling mechanism. Both steps in the Förster process occur with the

distance-dependent first-order rate constant $k_{Förster}$. The resulting $S_n$ species decay either by internal conversion to $S_1$ (rate constant $k_n = \tau_n^{-1}$) or by the formation of a CP state ($k_{CPn} = \tau_{CPn}^{-1}$). The electrons and holes produced by CP formation eventually recombine to repopulate the $S_0$ ground state (rate constant $k_{rec}$).

This kinetic model was implemented using the programme package Tenua 2.1[44], and details of the implementation are provided in Supplementary Note 7. The numerical solution provided an optimum fit, which is displayed in Fig. 7c. The resulting kinetic parameters are summarised in Supplementary Note 8. The TrCD signals at high (top panels) and low (bottom panels) laser fluence are displayed in terms of the absolute exciton number densities. As already pointed out above, the TrCD kinetics below 520 nm only monitor the $S_0$ ground state population and do not show contributions from excited states. Therefore, the experimentally measured values (blue circles) directly reflect the number density of $S_0$, which in this case makes the modelling procedure of the TrCD response much easier than for a transient absorption signal. An extensive kinetic analysis of the TrCD kinetics showed that the fast recovery of $S_0$ population at early times (left panels) is predominantly due to diffusive $S_1$−$S_1$ exciton annihilation ($S_1$ exciton number density shown as red lines), whereas FRET has only a minor contribution. For the lower initial exciton density, the rise of the $S_0$ population significantly slows down because of the second-order character of the diffusive process. For further illustration, simulations for the limiting cases (pure diffusive and pure FRET behaviour) are provided in Supplementary Note 9. Lastly, the TrCD experiments provide clear information on the efficiency of charge-pair formation for the highly excited $S_x$ and $S_n$ exciton states of c-PFBT. Taking the rate constants from the best fit, we obtain a yield of 9% for CP formation from the initially excited $S_x$ state and about 6% for the same process in the $S_n$ state accessed by the SSA processes, resulting in a total CP yield of 15%.

In summary, these broadband transient circular dichroism experiments in the UV−Vis range show that the ultrafast electronic response of chiral cholesteric c-PFBT copolymer thin films is of supramolecular origin. The TrCD signal of c-PFBT exclusively tracks population changes in the $S_0$ ground state (in contrast to the simultaneously recorded transient absorption) and allows for a particularly simple interpretation of the TrCD kinetics and the electronic relaxation processes in the film. Spatial mapping of the TrCD response allows quantifying local variations of the transient chiral response of these films and therefore complements steady-state CD and CPL imaging approaches.

## Methods

**Preparation of c-PFBT thin films.** Intrinsically chiral PFBT (c-PFBT) copolymer with a number average molar weight ($M_n$) of 7.46 kg mol$^{-1}$ and polydispersity (PDI) of 2.44 was prepared via Suzuki polycondensation from 2,7-bispinacoyl-9,9-bis((S)-3,7-dimethyloctyl)fluorene boronic ester, 3,6-dibromothiadiazole and 2-pinacoyl-9,9-bis((S)-3,7-dimethyloctyl)fluorene boronic ester and then purified afterwards[17,18]. A solution of c-PFBT in a chlorobenzene:chloroform mixture (1:9 by volume, concentration 7.5 mg ml$^{-1}$) was spin-coated on thoroughly cleaned 1 mm-thick borosilicate glass slides. The film thickness was controlled by adjustment of the spin-coating conditions using an optional preloading time of 30 s and rotation speeds in the range 500–4000 rpm. The films were then annealed in a nitrogen atmosphere at 150 °C for 15 min. The thickness of the polymer films was determined in a contactless fashion by picosecond ultrasonics[27,28] as $d = 0.25 \cdot \tau_a \cdot c_L$, where $\tau_a$ is the measured coherent acoustic phonon oscillation period (obtained from the ultrafast transient absorption kinetics averaged over the wavelength range 470−510 nm) and $c_L$ is the known longitudinal sound velocity of c-PFBT (2490 m s$^{-1}$)[29].

**Steady-state optical spectroscopy.** Steady-state absorption spectra of the thin films were measured using a Varian Cary 5000 spectrophotometer (slit width 0.5 nm). CD spectra were recorded on the same instrument using a home-built add-on employing polariser—achromatic quarter-wave plate combinations (Thorlabs WP25M-UB and AQWP05M-340 for the wavelength range 260−410 nm, Thorlabs LPVISE100-A and AQWP05M-580 for the spectral

region 400−700 nm). The CD spectrum was then obtained by taking the difference of two measurements, with the fast axis of the quarter-wave plate adjusted to either +45° or −45° with respect to the axis of the polariser. Steady-state PL and CPL spectra of the complete thin film samples were obtained as follows: The thin film was illuminated by light from a filtered (Schott UG 1, 3 mm) continuous-wave UV LED (Thorlabs M365LP1, 365 nm, FWHM 10 nm). Fluorescence emitted at right angle passed through a zero-order achromatic broadband quarter-wave plate and a broadband polariser (same combination as for the steady-state CD experiments) and was then focused into a fibre-optic cable connected to a spectrograph with a back-illuminated thermoelectrically cooled CCD detector (Avantes AvaSpec-Hero). CPL spectra were obtained by subtracting the emission spectra of two consecutive measurements with the polariser axis set at either 0° or 90° and the fast axis of the quarter-wave plate fixed at 45°.

**Steady-state circular dichroism microscopy.** CD microscopy images were recorded on an inverted microscope (Olympus IX71). Illumination of the c-PFBT sample was performed by a halogen lamp using a modified condenser setup featuring an additional polariser (Thorlabs LPVISE100-A) and an achromatic broadband quarter-wave plate (Thorlabs AQWP05M-580). The transmitted light was collected by a microscope objective (Olympus LCPLFL, 40×, NA 0.60) and sent through a bandpass filter (Thorlabs FB470-10, 470 nm centre wavelength, FWHM 10 nm). It then passed a 1:1 matched achromatic doublet pair ($f_1 = f_2 = 100$ mm, Thorlabs MAP10100100-A) and was divided by a non-polarising 50:50 beam splitter (Thorlabs CCM1-BS013/M). One part was detected by a CCD camera (PCO Sensicam QE) to obtain images with a diffraction-limited resolution of about 500 nm. The other part was focused by a quartz lens into an optical quartz fibre (600 μm core diameter) connected to a spectrograph, which was equipped with a back-illuminated thermoelectrically cooled CCD detector (Avantes AvaSpec-Hero) to record the intensity integrated over the entire field of view. Measurements for the light intensity of the c-PFBT sample ($I$) and a reference glass slide ($I_0$) were recorded with the fast axis of the quarter-wave plate set at either +45° or −45° with respect to the polariser axis. For comparison, conventional crossed-polariser images were obtained by removing the quarter-wave plate and introducing another polariser (Thorlabs LPVISE100-A) instead of the bandpass filter.

**Steady-state circularly polarised luminescence microscopy.** For CPL microscopy, the light at 365 nm (FWHM 14 nm) emitted from a 150 W xenon lamp/monochromator combination (Till Photonics Polychrome 5000) was coupled into the same inverted microscope by a quartz fibre to excite the c-PFBT sample through the microscope objective. The luminescence passed through a long-pass filter (440 nm) and an achromatic broadband quarter-wave plate (Thorlabs AQWP05M-580). The beam was then divided by a non-polarising 50:50 beam splitter (Thorlabs CCM1-BS013/M). One part passed through a polariser (Thorlabs LPVISE100-A) and was detected by the CCD camera to obtain PL images (diffraction-limited resolution ca. 500 nm). The other part of the PL passed through another polariser (Thorlabs LPVISE100-A) and was then detected by the same lens/fibre/spectrograph combination already mentioned above, to record spectra integrated over the entire field of view. CPL images and CPL spectra were obtained from two measurements, in which the fast axis of the quarter-wave plate was set at either +45° or −45° with respect to the polariser axis. The CPL images were more blurred and had less contrast than the CD images, likely because of the longer integration time (2 s vs. 1−10 ms, resulting in a larger influence of mechanical vibrations of the setup) and isotropic fluorescence scattered and reflected at domain boundaries of the film.

**Ultrafast broadband transient circular dichroism spectroscopy.** The transient circular dichroism experiment is based on amplified titanium:sapphire laser system running at 920 Hz (Coherent Libra USP-HE, 800 nm). The linear polarisation of the second harmonic pulses (400 nm, 50 fs pulse length, ca. 20 μJ pulse⁻¹) was converted into left-circular polarisation by means of a high-precision quarter-wave plate (B. Halle Nachfl.). The circularly polarised beam then traversed a DKDP Pockels cell (Eksma MP1), replacing the BBO-based Pockels cell used in an older design[24,25]. Switching from left-circular to right-circular polarisation was achieved by applying a pulsed half-wave voltage of 2500 V. The circularly polarised 400 nm seed pulses were focused into a translating CaF₂ plate (thickness 3 mm), generating a broadband multifilament UV−Vis supercontinuum (260−700 nm), which was then separated into a signal and a reference beam by means of a broadband beam splitter. At the thin film sample, the circularly polarised signal beam (beam diameter 50 μm) was crossed by the linearly polarised output of an optical parametric amplifier (Coherent OPerA Solo, pumped by the same titanium:sapphire laser system), generating 50 fs pump pulses at 320 or 450 nm (beam diameter 200 μm). The pump pulses were time-delayed by a motorised translation stage and mechanically chopped at half of the probe beam repetition frequency. To exclude any unwanted contributions of quadrupole-field interactions to the TrCD signal, the magic angle of 35.3° between the propagation directions of the pump and probe beams was employed[45,46]. Complementary experiments at a pump−probe crossing angle of 11°, i.e. for a nearly collinear arrangement typically used in transient absorption experiments, were almost identical and therefore suggest that contributions from quadrupole-field interactions for the c-PFBT sample are negligible. The signal and reference beams were sent into two separate spectrographs

where they were imaged onto 512-element silicon photodiode arrays, with the reference measurement allowing for a shot-to-shot correction of the supercontinuum fluctuations. The resulting transient circular dichroism signal $\Delta CD = \Delta\Delta OD = \Delta OD_L - \Delta OD_R$ was obtained from four consecutive laser shots (LCP and RCP, each with and without the pump beam) and averaged 3000 times for a given delay time. The time-resolution of the setup was ca. 100 fs. The typical measurement time to acquire a complete spectral data set ($\Delta OD_L$, $\Delta OD_R$ and $\Delta\Delta OD$) for 300 different delay times with the S/N ratio shown in Fig. 4a was about 60 min for the c-PFBT copolymer thin films studied here. Initial exciton number densities were determined from the measured laser fluence, obtained by a calibrated photodiode (Thorlabs S120VC) and a CCD-camera-based beam profiler (Visulux) using the known absorbance and thickness of the thin films.

**Ultrafast broadband transient absorption spectroscopy.** Additional transient absorption measurements were recorded with the setup mentioned above, however with the first quarter-wave plate and Pockels cell removed and the linearly polarised pump and probe beams crossing at an angle of about 11° at the sample with the polarisation planes of both beams set at the magic angle of 54.7° to guarantee anisotropy-free detection. For both the transient absorption and transient circular dichroism measurements, the c-PFBT thin film was kept inside an aluminium cell under a constant flow of dry nitrogen. This cell was mounted on two piezo stages (minimum step size 0.1 μm) for random movement within a quadratic plane ($2 \times 2$ mm²), which was normal to the propagation direction of the probe beam. For spatial mapping, a quadratic area of the thin film sample was probed in a random fashion (1 mm², $20 \times 20 = 400$ points with 50 μm step size in each direction).

**Time-correlated single-photon counting (TCSPC).** Transient fluorescence decays of the c-PFBT thin films were collected using TCSPC on a Horiba Jobin-Yvon TemPro system with 55 ps/channel[47]. Photoexcitation was performed at 1 MHz repetition frequency using a pulsed LED (Horiba Scientific NanoLED, 454 nm, FWHM 26 nm, pulse duration 1.1 ns, 3.2 pJ pulse⁻¹). Stray light of the excitation beam was removed by means of a long-pass filter (Schott GG495, thickness 3 mm). The response function of the setup was determined using a TiO₂ thin film sample. A deconvolution procedure was applied to extract the time constants and amplitudes from a biexponential fit.

**Kinetic analysis of transient circular dichroism data.** The kinetic scheme provided in Fig. 7b was implemented and numerically solved using the programme Tenua 2.1[44]. The rate constants in the mechanism were varied to arrive at an optimal description of the TrCD decay kinetics.

## Data availability
The data that support the findings of this study are available from the corresponding authors upon request.

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

## Acknowledgements
The setup for the transient fluorescence experiments was kindly made available by J. Arden-Jacob and K.H. Drexhage (ATTO-TEC GmbH Siegen). K. Oum and T. Lenzer thank the Deutsche Forschungsgemeinschaft (DFG) for financial support through grants OU 58/11-2 and LE 926/12-2.

## Author contributions
K.O. and T.L. conceived the project. M.M. prepared the thin film samples and performed the steady-state absorption experiments. M.M. and M.S. conducted the CD and CPL spectroscopy and microscopy experiments. K.O., T.L., and M.M. carried out the transient circular dichroism and transient absorption experiments and performed the data evaluation. M.S., K.O., and T.L. implemented the optical and electronic setups and the CD and CPL microscopy. M.J.C. and D.H.C. synthesised the copolymer samples. T.L., M.M. and K.O. analysed the spectroscopic results and wrote the paper.

## Funding

## Competing interests
The authors declare no competing interests.

## Additional information

**Peer Review Information** *Nature Communications* thanks, Stefan Meskers, and the other, anonymous, reviewer(s) for their contribution to the peer review of this work. Peer reviewer reports are available.

