## [Peer Review File · Nature Communications]

REVIEWER COMMENTS

Reviewer #1 (Remarks to the Author):

The study by Morgenroth, Scholz, Cho, Choi, Oum, and Lenzer takes on the time resolved optical activity in films of polyfluoroene type pi-conjugated polymer takes us into completely new territory. The measurement techniques employed are state-of-the-art and the results provide new and unprecedented insights into the optical properties of these kinds of films.

The analysis of the results is quite complicated and will certainly trigger discussion. Yet given the extreme novelty of the data obtained, the interpretation provided by the authors is reasonable and may well be a gambit in a scientific discussion that will likely continue in the coming years. I am happy to recommend publication in Nature Communications of this groundbreaking work. Below I have listed a few issues that the authors could consider before publication.

1) Page 4 Here the authors write “Therefore, we can safely conclude that contributions of linear dichroism (LD) and linear birefringence (LB) must be small.” I think this statement is incomplete. The fact that the signals do not change when rotating the sample indicates that there are no significant contributions stemming from a combination of linear anisotropies (LD and LB) in the sample in combination with any possible anisotropies in the spectroscopic setup. The anisotropies in the setup will of course not change when rotation the sample. These steady anisotropies combined with changing anisotropies in the sample may produce artificial circular dichroism signals of e.g. the type $LD_{\text{sample}} * LB'_{\text{setup}}$. These cross terms between anisotropies in the moving and static parts are expected to reveal themselves through changes in sign upon rotation of the sample. A combination of LD and LB' within the sample will not reveal itself in this way and constitutes a genuine contribution to the circular differential response of films with long range helical order.

2) The presentation of the CD measurements is not consistent and should preferably be harmonized. For instance, Fig. 1 show shows CD expressed as ellipticity in units of degrees, while Figure 5 show the transient circular dichroism as circular differential absorbance expressed in mOD. For clarity I would recommend expressing all circular dichroism as circular differential absorbance. This can also help people outside the field of optical activity to easier understand the results.

3) In the same vein, Fig. 2 shows results of measurements the circular polarization of luminescence (CPL). Here it is convention to label the vertical axis as $\Delta I = I_{\text{Left}} - I_{\text{Right}}$, i.e. the circular differential intensity ((a dimensionless quantity). ‘CPL’ usually refers in a qualitative sense, and is not used as an quantitative measure.

4) On page 13, the authors write “At low fluence, the S1 state gives rise to circularly polarized stimulated emission”. This is not completely clear to me. Naively, I would not expect much stimulated emission at low probe fluence. I guess that the author refer to low pump fluence, please clarify.

As a side step, in conventional spontaneous CPL measurements are limited by Poisson noise inherent in the photon counting process. This gives in very good approximation a standard error for g_{lum} equal to $err(g_{lum}) = 2/\sqrt{N}$ with N the number of photons counted. Here obviously the number of photons counted is less than the number of molecules that have been excited. Now for the new circularly polarized stimulated emission (CPSE) measurements, would the noise characteristics be different?

5) The modelling of the polarization transients worries me a little. The authors have just argued that the circular polarization arises most likely from supramolecular ordering. But, confusingly, in the modelling the authors seem to return, tacitly, to an interpretation where every photoexcitation contributes equally to the polarization transients. Yet the research by e.g. the group of Shaw Chen in Rochester on cholesteric liquid crystals doped with achiral fluorescent dye molecules has shown that the circular polarization in the fluorescence of the dye molecules induced by the supramolecular organization of the host depends on the position of the photoexcited dye molecule within the film. Here something a similar mechanism could be operative where the photoinduced polarization properties of an excitation would depend on its position within the film. Could the authors comment on this possibility? In any case it would be helpfully if the authors identify explicitly any relevant tacit assumptions in their modelling.

Stefan Meskers

Reviewer #2 (Remarks to the Author):

Chiral spectroscopy, in particular ultrafast CD absorption experiments, have long been searching for the needle in the haystack, trying to identify the tiny chiral response of small molecules that can be hidden under a plethora of achiral artifacts. In this paper, Morgenroth et al take a different approach. They investigate a material, whose huge circular dichroism and circular polarized luminescence not only make it interesting for potential applications (and more forgiving to optical imperfections of the set-up). The authors also demonstrate that ultrafast CD spectroscopy and microscopy can be important to better understand such materials and the mechanisms underlying their interesting optical properties.

For the thin copolymer films investigated in this study the combined results of CD and CPL microscopy and ultrafast CD spectroscopy provide convincing evidence for a supramolecular origin of the large dissymmetry factors. Particularly interesting in this respect is the interpretation of the time-resolved CD signals with contributions from 'stimulated CPL' and the absence of excited state circular dichroism. The model put forward on page 14 and in Figure 5a provides a qualitative explanation for the missing excited-state CD, and I wondered, if this model could be tested by doping the material with different chromophores. However, it is not fully clear to me why a change in 'in-plane and across-planes longer-range coupling' by some localized excitations would primarily weaken and not simply change the shape of the ground state CD. I also had to carefully read ref. 22 in order to understand why CPL can still be strong within this model, and the authors may want to add a few words of explanation (in a medium with very strong CD at the emission wavelength a linearly polarized photon becomes elliptically polarized).

In addition, I have a few technical questions:

The authors have checked the space-averaged CD signal for invariance under sample rotation, from which they conclude that linear birefringence and dichroism artifacts are small. Did they also check the invariance of a CD image under sample rotation? It might be, that linear birefringence contributes significantly to the spatial variations of the CD signal but averages out over a larger area. Likewise, it would be instructive to compare full spectra of 'bright-CD' and 'dark-CD' spots in the images.

Is there an explanation for the different resolution of the images a,b (CD) and c,d (CPL) in Figure 2? Are the two regions of the sample so different? Ideally, the same sections should have been scanned. Did the authors also record conventional crossed polarizer images and do they reveal the same structures and sample heterogeneity?

Finally, in the last section (pages 19-22), the transient data is fitted to a plausible, but by no means unique kinetic model. While the assumption that the CD signals are sensitive to ground-state population provides an important constraint, the data still seems to be overfitted, given that only two excitation densities have been analysed.

When these points can be addressed, this will be an exciting paper that introduces advanced novel methods for the characterization of materials with large supramolecular chirality. It will, however, require much higher sensitivity in the future to record similar spectro-temporal images of more ordinary molecular samples.

Reviewer #3 (Remarks to the Author):

The manuscript by Morgenroth et al. reports a chiroptical investigation of thin films of PFBT. The system is well-known and has been selected as proof-of-principle for the use of cutting-edge techniques such as transient ECD/CPL, ECD/CPL mapping, and, very notably, a combination thereof. This latter combination is unprecedented and deserves itself urgent publication. I definitely recommend publication, almost in its current version, provided that an expert of TrCD (that I am not) will also evaluate the manuscript.

Minor points:

- 1) Please avoid the use of “natural” ECD as a synonym of “single molecule ECD”. ECD signals from supramolecular entities are perfectly natural (in the sense they arise from the product of magnetic and electric moments). This is different from apparent ECD due to Bragg’s reflection.
- 2) Similarly, please avoid “magneto-electric coupling” as the source of “natural CD”, as the former term usually applies to static effects like in multiferroic materials.
- 3) I couldn’t find any reference for ECD/CPL mapping in the paper, though this technique (in its “steady state” version) is nowadays well established, see e.g. DOI 10.1021/acs.macromol.6b02590, 10.1039/c9nj02746g and references therein.
- 4) Related to the previous point, ECD/CPL mapping does not point at multiple aggregation species, as it is the case for some of the mentioned reports. Can the author comment on this fact at the top of page 10?
- 5) How can the authors exclude that the absorption bands > 500 nm are due to scattering, since circular selective scattering is invoked for long-wavelengths CD bands? They comment that “a similar weak feature is also observed in the ground-state CD spectrum”, which is however not really apparent in Figure 1.
- 6) I cannot agree with the statement “CPL is also thought to be a long-range supramolecular effect, likely resulting from circular intensity differential scattering”. CPL signals from chiral supramolecular assemblies, exactly like CD ones, are not necessarily related to CIDS, though CIDS may produce chiroptical signals both in absorption and emission.

Response to the reviewers' comments

(Note: Reviewers comments, reproduced verbatim, are in italic font, with central points highlighted in bold. Authors' replies are in normal font. Changes are highlighted in red colour.)

Authors' replies to the comments of Reviewer 1

*“The study by Morgenroth, Scholz, Cho, Choi, Oum, and Lenzer takes on the time resolved optical activity in films of polyfluorene type pi-conjugated polymer **takes us into completely new territory.** The measurement techniques employed are state-of-the-art and **the results provide new and unprecedented insights into the optical properties of these kinds of films.** The analysis of the results is quite complicated and will certainly trigger discussion. Yet given the **extreme novelty of the data obtained,** the interpretation provided by the authors is reasonable and may well be a gambit in a scientific discussion that will likely continue in the coming years. **I am happy to recommend publication in Nature Communications of this groundbreaking work.** Below I have listed a few issues that the authors could consider before publication.”*

Authors' reply: We thank reviewer 1 for the very positive overall evaluation of our study and take this as an incentive for further high-quality follow-up investigations on such chiral supramolecular systems, which will hopefully be of value for the whole community.

*“1) Page 4 Here the authors write “Therefore, we can safely conclude that contributions of linear dichroism (LD) and linear birefringence (LB) must be small.” **I think this statement is incomplete. The fact that the signals do not change when rotating the sample indicates that there are no significant contributions stemming from a combination of linear anisotropies (LD and LB) in the sample in combination with any possible anisotropies in the spectroscopic setup.** The anisotropies in the setup will of course not change when rotation the sample. These steady anisotropies combined with changing anisotropies in the sample may produce artificial circular dichroism signals of e.g. the type $LD_{sample} * LB'_{setup}$. **These cross terms between anisotropies in the moving and static parts are expected to reveal themselves through changes in sign upon rotation of the sample.** A combination of LD and LB' within the sample will not reveal itself in this way and constitutes a genuine contribution to the circular differential response of films with long range helical order.”*

Authors' reply: We fully agree with the reviewer, that this statement was way too short and therefore incomplete. On page 6, we thus replaced our original statement “Therefore, we can safely conclude that contributions of linear dichroism (LD) and linear birefringence (LB) must be small.” by a formulation very close to the one suggested by the reviewer:

*“... **Therefore, we conclude that there are no significant contributions resulting from a combination of linear anisotropies (i.e. linear dichroism and linear birefringence) in our sample in combination with any possible anisotropies in the spectroscopic setup. ...”***

*“2) **The presentation of the CD measurements is not consistent and should preferably be harmonized.** For instance, Fig. 1 show shows CD expressed as ellipticity in units of degrees, while Figure 5 show the transient circular dichroism as circular differential absorbance expressed in mOD. **For clarity I would recommend expressing all circular dichroism as circular differential***

absorbance. This can also help people outside the field of optical activity to easier understand the results.”

Authors’ reply: We agree with the reviewer that harmonizing the presentation of the CD experiments in terms of a common absorbance scale in all figures will help people to easier understand the results. Consequently, we now express all the steady-state CD values (originally given as ellipticity in units of degree) also as absorbance values, based on the well-known conversion formula $1 \text{ OD} = 32.982 \text{ deg}$. For the sake of a shorter terminology, we now refer in several instances to “left-circularly polarised” and “right-circularly polarised” simply as “L” and “R”, respectively, in addition to the already used abbreviations “LCP” and “RCP”. Otherwise, the labels on the y-axes would become too long. Changes have been made to the y-axes in **Figs. 1b-d, 2a (right side), 3a-d (right side) and Supplementary Fig. 1a-d (right side)**. Also, as a consequence, some formulations in the main text required adjustment:

page 2: “Broadband electronic circular dichroism spectroscopy, measuring the difference in optical density (OD) between left-circularly polarised (**LCP or shortly L**) and right-circularly polarised (**RCP or shortly R**) light (i.e. $\text{CD} = \text{OD}_L - \text{OD}_R$), is a powerful method ...”.

page 3: “... the transient CD signal as the difference in optical density for probing with LCP and RCP light ($\Delta\text{CD} = \Delta\text{OD} = \Delta\text{OD}_L - \Delta\text{OD}_R$). Alternatively, ...” and later on: “... and transient absorption (TA) signals ($\Delta\text{OD} = 0.5 \cdot [\Delta\text{OD}_L + \Delta\text{OD}_R]$) from four consecutive ...”

page 4: “... Steady-state absorption spectra recorded with left-circularly and right-circularly polarised light showed substantial differences (blue and red lines in the top panel of Fig. 1b), resulting in a strong CD response **with OD values of up to -0.3 ($= -10000 \text{ mdeg}$)** and a dissymmetry factor $g_{\text{abs}} = 2(\text{OD}_L - \text{OD}_R)/(\text{OD}_L + \text{OD}_R)$ approaching -0.5 ...”

page 10: “... **The shape** of the CD spectra integrated over all regions is quite similar, with maximum OD amplitudes between **-0.21 (-7000 mdeg) and -0.30 (-10000 mdeg)**, **as shown on the right side. ...**” and later on: “... The peak value of the CD spectrum in b is about 5% larger (-0.30 vs. -0.29 in OD). ... The peak value of the CD spectrum in d is about 17% smaller (-0.21 vs. -0.25 in OD), yet ...”

Note that in the text, we occasionally use both absorbance and ellipticity simultaneously to better serve researchers from both communities (time-resolved spectroscopy and optical activity).

“3) In the same vein, Fig. 2 shows results of measurements the circular polarization of luminescence (CPL). Here it is convention to label the vertical axis as $\Delta I = I_{\text{Left}} - I_{\text{Right}}$, i.e. the circular differential intensity (a dimensionless quantity). ‘CPL’ usually refers in a qualitative sense, and is not used as a quantitative measure.”

Authors’ reply: We thank the reviewer for pointing this out. We replaced “CPL” in the respective figures by “ $I_L - I_R$ ”, see **Fig. 1b** and **Fig. 2c and d**. Consequently, the text in the manuscript has been modified on page 4: “... In addition, the films displayed strongly polarised photoluminescence with a dissymmetry factor $g_{\text{lum}} = 2(I_L - I_R)/(I_L + I_R)$ of up to -0.4 (Fig. 1b, brown lines). ...”. We note that the measured PL intensity corresponds to a power per unit area in units W m^{-2} . Because we did not perform an absolute intensity measurement, we still keep the description “(arb. units)” on the y-axes. Also note that, according to editorial recommendations of *Nature Communications*, **for all microscope images in the paper (such as in Figs. 2 and 3) we have switched to a perceptually uniform colour scheme, which is also appropriate for colour-vision deficient people.**

“4) On page 13, the authors write “At low fluence, the S_1 state gives rise to circularly polarized stimulated emission”. This is not completely clear to me. Naively, I would not expect much stimulated emission at low probe fluence. I guess that the author refer to low pump fluence, please clarify.

As a side step, in conventional spontaneous CPL measurements are limited by Poisson noise inherent in the photon counting process. This gives in very good approximation a standard error for $glum$ equal to $err(glum) = 2/\sqrt{N}$ with N the number of photons counted. Here obviously the number of photons counted is less than the number of molecules that have been excited. Now for the new circularly polarized stimulated emission (CPSE) measurements, would the noise characteristics be different?”

Authors’ reply: Regarding the first half of the reviewer’s comment, “low fluence” indeed meant low pump fluence, as suggested by the reviewer. At low pump fluence, the S_1 exciton concentration is small, so S_1 – S_1 annihilation processes (bimolecular diffusive recombination and FRET) are slow. Therefore, the effective S_1 lifetime is longer, and one can observe a clearer (positive-going) CPSE feature in the TrCD spectrum shown in Fig. 4b due to the larger S_1 concentration at the time delay of 3 ps. At high pump fluence, the S_1 – S_1 annihilation processes are much faster, so already at 3 ps a considerable amount of S_1 population is depleted because of the annihilation processes (cf. the S_1 populations shown in Fig. 7c for “low” and “high” exciton number densities). In contrast, the fluence of the supercontinuum probe beam does not change the shape of the resulting transient spectra, it only has an influence on the signal-to-noise ratio.

To clarify this point, starting from page 14, we therefore changed “fluence” to “**pump fluence**”. We left the original explanation of the effect as is:

“... Figure 4b contains **spectra for different pump fluences**, with initial exciton number densities $N_0(S_x)$ in the range $5.7 \times 10^{17} - 1.2 \times 10^{19} \text{ cm}^{-3}$ at the fixed pump–probe delay time of 3 ps. ... In contrast, above 500 nm there is a change in sign: At low **pump** fluence, there is a negative band below 580 nm, which we assign to stimulated emission from S_1 , whereas at high **pump** fluence there is absorption in the entire 500–650 nm range. ... This process becomes faster at high **pump** fluence, and thus high initial exciton number densities in the polymer^{18,33}, as described by our kinetic modelling below. Therefore, under **such** high-fluence conditions, the S_1 population is already considerably depleted at 3 ps. ... The TrCD signal amplitude depends linearly on **the pump** fluence (Fig. 4c, black open circles and fit line). ... At low **pump** fluence, the S_1 state gives rise to circularly polarised stimulated emission (CPSE), i.e. a difference in the induced S_1 emission for left-circularly and right-circularly polarised probing. ... This overlapping CPSE band will therefore lead to an apparent broadening of the band toward higher wavelengths at low **pump** fluence. At the same time, it compensates for the negative contribution above 500 nm, which is the only contribution in this spectral range at high **pump** fluence (cf. the blue TrCD spectrum above 500 nm in the bottom panel of Fig. 4b) and is due to circular selective scattering. **At high pump fluence**, CPSE at 3 ps will be largely absent because of fast depletion of the S_1 state due to singlet–singlet annihilation, leading to an apparent blue-shift of the band in the TrCD spectrum¹⁸. ...”

Regarding noise characteristics of the CPSE (second half of the comment): Measurements of spontaneous emission (as any other statistical process, such as radioactive decay) are indeed limited by Poisson noise, as e.g. in typical setups for time-correlated single-photon counting (TCSPC). However, CPSE, and stimulated emission (SE) in general, is not a spontaneous process,

but triggered by an incoming probe photon, which induces an $S_1 \rightarrow S_0$ transition involving the emission of an “identical” photon replica (same wavelength, direction, phase, ...). In fact, the stimulating field of the probe pulse is quite intense, and therefore we are far away from a single-photon counting situation. Therefore, Poisson noise is not relevant for the noise characteristics of our transient absorption / stimulated emission experiments. Instead, the noise arising from the (stimulating) laser field is important. This “laser noise” stems from pulse-to-pulse fluctuations in the supercontinuum probe beam spectrum (both in intensity and spatial distribution). In our setup, the influence of the laser noise is minimised by recording the supercontinuum fluctuations for the same laser shot in a separate (“identical”) spectrograph and correcting for these. Full details of the procedure, including a thorough combined experimental and theoretical investigation of the noise floor in such experiments, can be found in the paper of Dobryakov *et al.* (*Rev. Sci. Instrum.* **81**, 113106 (2010)). The resulting expression for the confidence interval of a ΔOD measurement in that paper is $C \cdot \sigma_r/n^{1/2}$, where C is a constant depending on the procedure how the shot-to-shot referencing is performed (e.g. same shot or consecutive shots), σ_r is the variance of the statistical experimental data, and n is the number of pump-probe cycles. We newly included a comment on this noise aspect in the paragraph on page 15 (middle):

“... Note that the noise characteristics of the transient absorption and TrCD signals (and thus also the CPSE contribution) is governed by “laser noise”, i.e. shot-to-shot fluctuations of the intensity and spatial profile of the laser pulses⁴⁰. In contrast, measurements of spontaneous CPL are limited by Poisson noise.”

“5) The modelling of the polarization transients worries me a little. The authors have just argued that the circular polarization arises most likely from supramolecular ordering. But, confusingly, in the modelling the authors seem to return, tacitly, to an interpretation where every photoexcitation contributes equally to the polarization transients. Yet the research by e.g. the group of Shaw Chen in Rochester on cholesteric liquid crystals doped with achiral fluorescent dye molecules has shown that the circular polarization in the fluorescence of the dye molecules induced by the supramolecular organization of the host depends on the position of the photoexcited dye molecule within the film. Here something a similar mechanism could be operative where the photoinduced polarization properties of an excitation would depend on its position within the film. Could the authors comment on this possibility? In any case it would be helpfully if the authors identify explicitly any relevant tacit assumptions in their modelling.”

Authors’ reply: This is an important point raised by the reviewer and was probably not emphasised sufficiently by us in the original manuscript (but could have been already based on the data presented). We think that a “supramolecular” response based on a “molecular” excitation is not a contradiction. Regarding this point, there are two important findings in our experiments:

(1) First of all, panels b (bottom) and panel c of Fig. 4 (black points and line) show that for a given film thickness, the TrCD signal depends linearly on the pump fluence. That is, every photoexcitation indeed contributes equally to the polarization transient, as stated by the reviewer. This is a cornerstone of our kinetic modelling in Fig. 7, because otherwise we would not be allowed to relate the TrCD signal amplitude to the S_0 concentration.

(2) Secondly, comparing Fig. 4a and 5b, we see that for two different thicknesses, and keeping the same transient absorption amplitude, the TrCD signal amplitude of the thinner film is much smaller than for the thicker film. This is underlined by the “dynamic dissymmetry factor” $g_{\text{TrCD}}(t)$ which is about twice as large for the thicker film (cf. Fig. 5d). This means that, although both films have

about the same number density of photoexcited species (same bleach amplitude in the transient absorption response), the TrCD response of the thicker film is much bigger.

These two findings suggest that the TrCD signal, in contrast to transient absorption, scales with an additional factor, which depends on the film thickness, and this factor is obviously considerably larger for thicker films. Still, this constant factor does not affect the linearity of the TrCD signal.

Photoexcitation of the polymer always involves the promotion of a single electron residing on an individual chromophore unit (second law of photochemistry). This electron afterwards either stays on the same chromophore (molecular excitation) or gives rise to a collective excitation. The latter process is related to a supramolecular picture, such as an exciton-coupling mechanism. Irrespective of where this electron ends up, this process (the pump step in a pump-probe experiment) “knocks out” a ground state chromophore. It is this loss of S_0 population, which is seen as a bleach feature both in our TrCD (and also the transient absorption) spectra. Therefore, we think that a linear dependence of the TrCD signal on the pump fluence and a “supramolecular amplification” of the TrCD signal do not contradict each other.

The second part of the reviewer’s comment refers to the physical origin of the supramolecular TrCD response. There are two central questions remaining here:

(1) What is the specific “supramolecular amplification” mechanism in the TrCD signal?

As discussed in the literature (e.g. Lakhwani & Meskers, *J. Phys. Chem. A* **116**, 1121 (2012), Ou & Chen, *J. Phys. Chem. B* **124**, 679 (2020)), the steady-state supramolecular CD response of these systems can be described within the formalism developed of Good & Karali (based on the Maugin-Oseen-DeVries approach) or molecular exciton coupling theory. As outlined in the first reference, special care has to be taken, which regime of layer thickness is considered. As these approaches appear to be quite successful in describing the shape of the steady-state CD spectra of cholesteric copolymer thin films, we think that it is adequate to interpret the TrCD bleach component simply as a corresponding loss of supramolecular S_0 signal due to bleached chromophores.

(2) Why is there no TrCD signal originating from the excited state species (S_1)?

Our tentative answer in our original manuscript was based on the simple scheme in Fig. 5a. As mentioned above, any supramolecular CD signal requires some sort of coupling mechanism between the different chromophores. The chiral supramolecular arrangement of S_0 species stays largely intact: Only about 10% of S_0 is “knocked out” (that is the population loss due to photoexcitation, which produces the bleach feature). In contrast, the resulting 10% excited-state species are widely distributed. We think that such a quite sparse statistical distribution of excited chromophores is not able to support a sizeable contribution of S_1 excited-state CD to the supramolecular TrCD response, i.e. the photoinduced polarisation properties of the excited S_1 species are therefore too weak (meaning there is only weak “single-molecule” chirality left).

The group of Shaw Chen in Rochester thoroughly studied cholesteric liquid crystals doped with achiral fluorescent emitters, such as laser dye molecules doped into chiral nematic films. For films showing selective reflection, outside the resonance region, the degree of circular polarisation was found to originate in linearly polarised luminescence of quasi-nematic layers with subsequent circular polarisation by the remaining film layer (Katsis et al., *Liq. Cryst.* **26**, 181 (1999)). Inside the resonance band region, a high degree of circular polarisation accompanied by reversal in handedness at the band edge was reported (Chen et al., *Nature* **307**, 506 (1999)). We think that for our multidomain cholesteric films the “outside resonance case” provides one reasonable mechanism for the generation of the CPSE features. Others groups suggest circular selective

scattering (Di Nuzzo et al., *ACS Nano* **11**, 12713 (2017), Sharma et al., *J. Phys. Chem. Lett.* **10**, 7547 (2019)) as another reasonable mechanism for the strong circular polarisation of emission. While these mechanisms are likely operative for CPL and CPSE, we have difficulties to imagine, how they could contribute to excited-state CD of the S_1 species. In fact, as mentioned above, we experimentally do not have clear indications for excited-state CD features in the TrCD signals, there are only CD bleach and CPSE bands.

Based on the discussion above, we modified the paper in several places to address the points of the reviewer:

On p. 18, we added a statement regarding the issue of molecular photoexcitation and supramolecular response in the TrCD signal:

“The TrCD signal for a given film thickness is still linearly dependent on the pump fluence, i.e. it scales with the number of initially excited polymer units (cf. Fig. 4b and c). The supramolecular nature of the transient chiral response manifests itself in terms of an additional “constant scaling factor” which increases the TrCD signal, that means it becomes much bigger for thicker films.”

and later, on pp. 20/21 in the modelling section:

“Also, the TrCD signal is linearly dependent on the S_0 concentration for a given film thickness (cf. Fig. 4b and c). The supramolecular nature of the TrCD response becomes evident for different film thicknesses, where thicker films show much larger TrCD signals, which however still increase linearly with the S_0 concentration.”

Regarding the mechanism for CPL/CPSE we reformulated on page 17 (now citing also the work of Shaw et al. mentioned by the reviewer):

“... As already mentioned, CPL is also thought to be a long-range supramolecular effect, likely resulting from processes such as circular intensity differential scattering^{22,36} and linearly polarised luminescence of quasi-nematic layers with subsequent circular polarisation by the remaining film layer⁴¹. Therefore, the CPSE contribution in the TrCD spectra, i.e. directed S_1 emission induced by the probe beam of the TrCD experiment, should be based on similar effects as the CPL signal. ...”

Authors' replies to the comments of Reviewer 2

1) “Chiral spectroscopy, in particular ultrafast CD absorption experiments, have long been searching for the needle in the haystack, trying to identify the tiny chiral response of small molecules that can be hidden under a plethora of achiral artifacts. **In this paper, Morgenroth et al take a different approach. They investigate a material, whose huge circular dichroism and circular polarized luminescence not only make it interesting for potential applications (and more forgiving to optical imperfections of the set-up). The authors also demonstrate that ultrafast CD spectroscopy and microscopy can be important to better understand such materials and the mechanisms underlying their interesting optical properties.**

For the thin copolymer films investigated in this study the combined results of CD and CPL microscopy and ultrafast CD spectroscopy provide convincing evidence for a supramolecular origin of the large dissymmetry factors. Particularly interesting in this respect is the interpretation of the time-resolved CD signals with contributions from ‘stimulated CPL’ and the absence of excited state circular dichroism. ...

When these points can be addressed, this will be an exciting paper that introduces advanced novel methods for the characterization of materials with large supramolecular chirality. It will, however, require much higher sensitivity in the future to record similar spectro-temporal images of more ordinary molecular samples.”

Authors’ reply: We highly appreciate the very positive evaluation of our manuscript by reviewer 2. Indeed, we are currently working on further sensitivity improvements of the TrCD setup in order to apply it to systems with a considerably smaller transient circular dichroism response.

2) *“The model put forward on page 14 and in Figure 5a provides a qualitative explanation for the missing excited-state CD, and I wondered, if this model could be tested by doping the material with different chromophores. However, it is not fully clear to me why a change in ‘in-plane and across-planes longer-range coupling’ by some localized excitations would primarily weaken and not simply change the shape of the ground state CD.”*

Authors’ reply: This remark of reviewer 2 is related to point 5 (second part) of reviewer 1, also commenting on the physical origin of the TrCD response. We believe, that our sentence in the original manuscript, which reviewer 2 refers to, was originally not formulated clearly. First of all, we meant that the main effect is the missing S_0 population due to photoexcitation, which leads to a “CD bleach” feature: The probe beam therefore “sees” about 10% less S_0 chromophores. Secondly, the photoexcited chromophores could indeed modify the ground state response a bit. Reviewer 2 is totally right that this could also change the shape of the S_0 ground state CD, not only weaken it, by changing the change in in-plane and across-planes longer-range couplings. Our TrCD experiments however do not find clear indications for such spectral changes. The signal looks just like an inverted steady-state CD spectrum. For clarification, we therefore reformulated the sentence on page 17 as follows:

“... One can well understand that photoexcitation makes about 10% of the S_0 chromophores inaccessible for the probe beam, which leads to the observed “CD bleach” feature in the TrCD spectra. In addition, the different electronic properties of the excited species will somewhat modify the in-plane and across-planes longer-range coupling between the electric and magnetic transition dipole moments of c-PFBT in the S_0 state. However, this effect appears to be weak, as there is a close resemblance of the TrCD spectrum to the inverted steady-state CD spectrum. ...”

Reviewer 2 suggests to dope the material with different achiral chromophores. This way, one would be able to check if the chiral supramolecular environment would induce circularly polarised luminescence (CPL) or circularly polarised stimulated emission (CPSE) of the intrinsically achiral guest chromophores. Indeed, such an experiment is a worthwhile target for future work. However, we have difficulties to see, how such a doping experiment would help to interpret the missing excited-state CD in the TrCD experiment on c-PFBT. Adding such chromophores at typically low doping concentrations (a few percent) would lead to widely distributed molecules, which could not establish some sort of supramolecular coupling among each other, to be later on probed by TrCD. That was exactly the message contained in Fig. 5a, namely to illustrate that the excited chromophores are then simply too far apart to support a supramolecular response. One would therefore expect a negative result of such an experiment. On the other hand, high doping concentrations would severely disturb the cholesteric arrangement of the c-PFBT layers, so any result from such an experiment would be difficult to interpret. To test our model, a separate detailed study of the expected excited-state CD response, possibly within the framework of the Good-

Karali theory or some exciton-coupling approach, could be helpful, but this is beyond the scope of our paper. Currently, the theoretical “machinery” to treat such an excited-state supramolecular interaction does not exist either.

3) *“I also had to carefully read ref. 22 in order to understand why CPL can still be strong within this model, and the authors may want to add a few words of explanation (in a medium with very strong CD at the emission wavelength a linearly polarized photon becomes elliptically polarized).”*

Authors’ reply: This comment of reviewer 2 refers to the similar point 5 (second part) raised by reviewer 1, regarding the physical origin of the supramolecular CPL, CPSE and TrCD signals. As requested by reviewer 2 (and reviewer 1), we added a remark regarding the most likely supramolecular CPL (and CPSE) mechanisms on page 17, also now referring to the work of the Chen group at the University of Rochester on this topic:

“... As already mentioned, CPL is also thought to be a long-range supramolecular effect, likely resulting from processes such as circular intensity differential scattering^{22,36} and linearly polarised luminescence of quasi-nematic layers with subsequent circular polarisation by the remaining film layer⁴¹. Therefore, the CPSE contribution in the TrCD spectra, i.e. directed S₁ emission induced by the probe beam of the TrCD experiment, should be based on similar effects as the CPL signal. ...”

4) *“In addition, I have a few technical questions:*

The authors have checked the space-averaged CD signal for invariance under sample rotation, from which they conclude that linear birefringence and dichroism artifacts are small. Did they also check the invariance of a CD image under sample rotation? It might be, that linear birefringence contributes significantly to the spatial variations of the CD signal but averages out over a larger area.

Authors’ reply: We thank reviewer 2 for this valuable experimental suggestion, leading to what we think are quite beautiful experimental results. We checked the invariance of the CD images under sample rotation and flipping. For illustration, we have added the two new Figs. S2 and S3 in the Supporting Information, containing CD images of a c-PFBT thin film rotated in 90° steps (and also flipped), g_{abs} distributions and spectra (OD_L, OD_R, CD, g_{abs}) integrated over the whole field of view, including some additional text. The figures nicely illustrate that there are indeed no local birefringence effects on micrometre length scales, which could possibly average out over macroscopic length scales. In the main manuscript, we added on page 8 a short reference to this new part in the Supporting Information:

“... We also checked the invariance of the CD images upon sample rotation and flipping, as illustrated in Supplementary Figs. 2 and 3, which proved that there are also no significant contributions on micrometre length scales resulting from a combination of linear dichroism and linear birefringence in our sample in combination with any possible anisotropies of the CD microscopy setup.”

5) ***“Likewise, it would be instructive to compare full spectra of ‘bright-CD’ and ‘dark-CD’ spots in the images.”***

Authors’ reply: Reviewer 2 refers to spatially resolved CD and fluorescence spectra, such as for the centre regions of the violet and yellowish spots (islands) in the CD and CPL images of Fig. 2 (please note the new type of colour map according to editorial recommendations), which belong to regions of weaker and stronger CD/CPL, respectively. It would be indeed interesting to see, if CD/CPL spectra of the bright and dark spots are different or not. However, our current microscope setup is not capable of supporting such a spatially resolved detection of CD/CPL spectra. The spectra, that we collect with our spectrometer at the exit port of the microscope, are integrating over the complete field of view. Recording spatially resolved CD spectra of micrometre-sized spots in the images would require a conceptually different approach using pointwise illumination employing a circularly polarised white-light laser source in some sort of modified confocal microscopy setup. Similarly, in the case of CPL detection, one would need a confocal fluorescence microscope, with localised fluorescence excitation by a laser source combined either with a galvanometer scanner or a piezo-based stage, and integration of the fluorescence signal of that spot over a longer period of time. Unfortunately, such microscope setups are currently not at our disposal and would need to be developed.

6) ***“Is there an explanation for the different resolution of the images a,b (CD) and c,d (CPL) in Figure 2? Are the two regions of the sample so different? Ideally, the same sections should have been scanned.”***

Authors’ reply: Indeed, the CPL images (Fig. 2c,d) are more blurred and have less contrast than the CD images (Fig. 2a,b). This is due to general technical limitations of wide-field fluorescence microscopy, specifically the following issues:

- (1) In the case of CPL imaging, fluorescence photons are emitted isotropically from each point in the field of view. Due to e.g. domain boundaries in the film, fluorescence photons “initially going sideways” can exit from locations adjacent to the originally excited spot due to reflection and scattering. This leads to blurring of the image. Because the films investigated are thin (240 nm or less), fluorescence emerging from above or below the focal plane, which could also increase blurring, is probably a minor issue. In contrast, for doing wide-field CD imaging (transmission measurements), light from a halogen lamp is sent through a well-aligned condenser arrangement optimised and the rays pass straight through the sample and are correctly coupled into the microscope objective.
- (2) Another well-known difficulty lies in the quite long CCD exposure times (typically a few seconds) required to record the fluorescence signal, which makes CPL imaging more susceptible to small vibrations of the microscope setup. In contrast, for a CD image (high-light situation) the required CCD exposure time is only 1–10 ms. Also, in CD imaging the lamp intensity can be much more easily matched to use the full 12 Bit A/D conversion range of the CCD camera (PCO Sensicam QE), which provides a better dynamic range.
- (3) Related to (2), at long exposure times, weak autofluorescence from optical elements, such as the glass substrate and the bandpass filter required for filtering out the excitation light may result in a background emission, which will further reduce the contrast of the CPL image. Dark current of the CCD may also play a role at long exposure times and reduce the contrast.

Therefore, the inferior quality of the CPL images compared with the CD images is not unexpected. Still, the CPL images are very valuable, because they contain sufficient detail to identify similar structures as in the CD images (cf. Fig. 2). Island-type structures of comparable size are visible in both images and the g_{abs} and g_{lum} distributions are consistent. For the example in Fig. 2, the CPL and CD images were indeed measured in different regions of the same film, but the film employed is homogeneous enough so that imaging of different areas provides consistent results.

We added a remark regarding this point in the Methods section on page 26:

“... The CPL images were more blurred and had less contrast than the CD images, likely because of the longer integration time (2 s vs. 1–10 ms, resulting in a larger influence of mechanical vibrations of the setup) and isotropic fluorescence scattered and reflected at domain boundaries of the film.”

7) “*Did the authors also record conventional crossed polarizer images and do they reveal the same structures and sample heterogeneity?*”

Authors’ reply: Yes, conventional crossed polariser images were also recorded. Indeed, they reveal the same structures and sample heterogeneity. Of course, a crossed-polariser image carries much less information than a CD or g_{abs} image. The crossed-polariser image only confirms that a material between the polarisers changes the polarisation and gives some information regarding the dimension of the polarisation-changing structures. We added a set of crossed-polariser images in the **new Sec. 2 of the Supporting Information (new Supplementary Fig. 4)** including a short description of the images.

On page 8, we added: “... In addition, the structures revealed in the CD images are very similar to the structures observed using a conventional crossed-polariser arrangement (Supplementary Fig. 4).”

In the Methods section on page 26 we added: “... For comparison, conventional crossed-polariser images were obtained by removing the quarter-wave plate and introducing another polariser (Thorlabs LPVISE100-A) instead of the bandpass filter.”

8) “*Finally, in the last section (pages 19-22), the transient data is fitted to a plausible, but by no means unique kinetic model. While the assumption that the CD signals are sensitive to ground-state population provides an important constraint, the data still seems to be overfitted, given that only two excitation densities have been analysed.*”

Authors’ reply: We agree with reviewer 3 that a larger number of parameters enters the kinetic model to fit the TrCD signal. However, a large part of these parameters is either determined independently (and therefore can be kept constant) or is very well defined. We believe that our original explanation in the Supporting Information (“Kinetic simulations for two limiting cases”) was simply too short. We therefore added a more extended discussion regarding this point at the end of that section:

“A larger number of parameters enters the kinetic model (Supplementary Table 1), so we would like to finally comment on the uniqueness of the fit and possible concerns regarding overfitting the TrCD data. We note that the majority of parameters is either determined independently (and therefore can be kept fixed) or is very well defined, as summarised by the following points:

1) Because the TrCD kinetics is only sensitive to the concentration of the S_0 population (a single species), there are no ambiguities regarding relative contributions of different overlapping CD-

active species with different species-associated CD values. The TrCD kinetics can be therefore taken as a direct measure of the time-dependent change of the S_0 number density.

- 2) We employed three different initial exciton number densities. In addition to the two initial number densities $N_{0,low}(S_x)$ and $N_{0,high}(S_x)$ in the TrCD experiments, there is also the “very low number density condition” covered by the TCSPC experiment (Supplementary Fig. 8) which directly provides the rate constant k_1 for the decay of the S_1 state (in the absence of singlet–singlet annihilation processes), see step S1. k_1 is therefore fixed.
- 3) The initial exciton number densities of the TrCD experiments and the TCSPC experiment were independently determined and are therefore also fixed.
- 4) In addition, the transient absorption kinetics in Supplementary Fig. 7 provides the total rate constant $k_{x,total}$ for the decay of the initially populated S_x state, which decays via two parallel channels. This value poses a strict limit on the sum of the two rate constants for the formation of the S_1 state (k_x) and the formation of the charge pair state (k_{CPx}). i.e. $k_{x,total} = k_x + k_{CPx}$.
- 5) The ratio of the rate constants k_x and k_{CPx} (steps S1 and S2) and the ratio of the rate constants k_n and k_{CPn} (steps S5 and S6) are also well defined by the experimentally observed yield for the long-lived CP state, which can be directly determined from the “incomplete recovery” of the TrCD kinetics. Only 85% of the S_0 population is recovered at about 500 ps, see Supplementary Figs. 9 and 10 (top right panel, blue line), so only 15% of the initially excited c-PFBT population ends up in the CP state (green line). The rate constant $k_{x,total}$ was measured independently. Therefore, k_x and k_{CPx} are well defined, with $k_x \gg k_{CPx}$. k_n and k_{CPn} are also well defined: Because the total yield of the CP state is only 15%, k_n must be considerably larger than k_{CPn} . In addition, the sum $k_{n,total} = k_n + k_{CPn}$ must be large, because the singlet–singlet annihilation processes would otherwise be too slow, and it would not be possible to fit the fast rise of the TrCD signal at early times.
- 6) As demonstrated by the systematic parameter variations in Supplementary Figs. 9 and 10, the dependence of the S_0 recovery on the different initial exciton number densities is governed by the competition of bimolecular diffusive SSA (step S3) and FRET (steps S14a and S14b), which leads to accurate values for the SSA rate constants k_{diff} and k_F .
- 7) The final electron–hole recombination step S7 has only a very weak influence on time scales up to 500 ps, as shown by the green lines in Supplementary Figs. 9 and 10 (right panels), and therefore the value for k_{rec} should only be taken as a rough estimate.

We therefore conclude that all parameters entering the kinetic model are well defined and provide detailed kinetic information regarding the individual processes involved.”

Authors’ replies to the comments of Reviewer 3

“The manuscript by Morgenroth et al. reports a chiroptical investigation of thin films of PFBT. The system is well-known and has been selected as proof-of-principle for the use of cutting-edge techniques such as transient ECD/CPL, ECD/CPL mapping, and, very notably, a combination thereof. This latter combination is unprecedented and deserves itself urgent publication. I definitely recommend publication, almost in its current version, provided that an expert of TrCD (that I am not) will also evaluate the manuscript.”

Authors’ reply: We thank reviewer 3 for the very favourable comments on our study.

“Minor points: 1) Please avoid the use of “natural” ECD as a synonym of “single molecule ECD”. ECD signals from supramolecular entities are perfectly natural (in the sense they arise from the product of magnetic and electric moments). This is different from apparent ECD due to Bragg’s reflection.”

Authors’ reply: We completely agree with reviewer 3. The term “natural” has been widely used in the CD and CPL literature, but is indeed misleading. Consequently, we changed the term “natural” by “single molecule” throughout the text:

page 3: “... (i.e. **single-molecule** optical activity)^{21,25}. Our experiments presented here, ...”

page 6: “... In contrast, **single-molecule circular** dichroism, where coupling between ...”

page 18: “... arguments against a mechanism based on **single-molecule** circular dichroism, ...”

“2) Similarly, please avoid “magneto-electric coupling” as the source of “natural CD”, as the former term usually applies to static effects like in multiferroic materials.”

Authors’ reply: We thank reviewer 3 for finding this incorrect term. We changed the manuscript accordingly:

page 3: “... who assigned the origin of the strong chiroptical response of c-PFBT to **the coupling of molecular magnetic and electric moments** (i.e. **single-molecule** optical activity)^{21,25}. ...”

“3) I couldn’t find any reference for ECD/CPL mapping in the paper, though this technique (in its “steady state” version) is nowadays well established, see e.g. DOI 10.1021/acs.macromol.6b02590, 10.1039/c9nj02746g and references therein.”

Authors’ reply: Indeed, relevant references regarding CD/CPL mapping techniques were missing. As suggested by reviewer 3, we therefore included some of the seminal studies: Maestre & Katz, Claborn et al., Savoini et al., Zinna et al. and Albano et al. on page 7:

“... **Several powerful setups for steady-state CD imaging have been established so far, starting from the early work of Maestre and Katz³⁰. These include circular dichroism imaging based on a wide-field microscope featuring illumination through a combination of an interference filter, a polariser and a tunable quarter-wave retarder³¹, scanning CD optical microscopy using a polariser and a photoelastic modulator³², CD microscopy with discretely modulated circular polarization³³, as well as scanning UV-Vis circular dichroism experiments employing highly collimated synchrotron radiation^{34,35}. Our specific implementation based on wide-field microscopy ...”**

“4) Related to the previous point, ECD/CPL mapping does not point at multiple aggregation species, as it is the case for some of the mentioned reports. Can the author comment on this fact at the top of page 10?”

Authors’ reply: We thank reviewer 3 for pointing this out. In the study of Albano et al. (*New J. Chem.* **43**, 14584 (2019)) drop casting or prolonged solvent annealing resulted in the formation of chiral oligothiophene films with CD responses, which were invariant upon flipping and originated from a pure CD_{iso} contribution. Spin-coating however resulted in films with a large LDLB contribution and a weaker CD_{iso} contribution. For both aggregation pathways, thermal treatment was not required to reach a chiral response and also did not improve it. Our polyfluorene-based copolymer thin films in general behave differently. Homogeneously coated areas are uniform and, as reviewer 3 mentioned, there are no clear indications for multiple aggregation pathways in our films. It may however be the case that some of the smaller changes we see in the CD images of

Fig. 3 in “non-optimally” spin-coated regions at the edge of the sample may be connected with slightly different aggregation pathways due to “non-optimal” spin-coating and/or annealing conditions. As recommended by reviewer 3, we added a short comment on page 11 regarding this issue:

“... It could well be that the different appearance in less homogeneous regions points toward different aggregation pathways, as previously suggested for oligothiophene films^{34,35}. Still, the thin film regions in the centre of the substrate and at intermediate distances are very uniform and also microscopically homogeneous, with no clear indications for different aggregation pathways.”

Again, please note that for all microscope images in the paper (such as in Figs. 2 and 3) we have switched to a perceptually uniform colour scheme, which is also appropriate for colour-vision deficient people.

“5) How can the authors exclude that the absorption bands > 500 nm are due to scattering, since circular selective scattering is invoked for long-wavelengths CD bands? They comment that “a similar weak feature is also observed in the ground-state CD spectrum”, which is however not really apparent in Figure 1.”

Authors’ reply: Reviewer 3 comments here on our discussion of the spectral region above 500 nm in the transient absorption spectra of Fig. 4. In each of the two transient absorption spectra (i.e. the blue lines for LCP and the red lines for RCP detection), those scattering contributions, which are independent of the polarisation of light will cancel out, because the signal is a difference measurement (signal pumped minus signal unpumped). Note that the largest part of the signal above 530 nm arises from transient absorption of the S₁ excitons and the CP state, and these LCP and RCP signals also cancel each other, because these “low-concentration” species are not capable of displaying a large supramolecular signal, as explained in the paper. The contribution remaining in the TrCD signal is therefore that for circular selective scattering, i.e. the difference in scattering for LCP and RCP light, plus a CPSE contribution. A comparison of the blue and red lines shows that the red line (RCP) is slightly higher than the blue line (LCP). This difference therefore shows up in the TrCD signal (green) as a tiny negative signal above 530 nm.

For comparison, we show below a “zoom-in” of Fig. 1b. It contains the steady-state CD spectrum in green and an extra black reference line at OD_L-OD_R = 0. One can observe that the CD spectrum above 530 nm is slightly above zero, which we interpret as an indication for weak circular selective scattering, because there are no S₀ absorption bands of the copolymer in that region. This weak steady-state circular selective scattering signal is consistent in amplitude with that of the TrCD experiment.

“6) I cannot agree with the statement “CPL is also thought to be a long-range supramolecular effect, likely resulting from circular intensity differential scattering”. CPL signals from chiral supramolecular assemblies, exactly like CD ones, are not necessarily related to CIDS, though CIDS may produce chiroptical signals both in absorption and emission.”

Authors’ reply: This comment of reviewer 3 refers to similar points raised by the other two reviewers, e.g. point 5 of reviewer 1 (second part), regarding the physical origin of the supramolecular CPL signal. Reviewer 3 is completely right that there are also other mechanisms beside CIDS. As explained in the reply to reviewer 1, there can be also other mechanisms, for instance the one proposed by Katsis et al., that there is linearly polarised luminescence emitted from the quasi-nematic layers with subsequent circular polarisation by the remaining film layer (Katsis et al., *Liq. Cryst.* **26**, 181 (1999)), meaning that the residual film material acts itself as a quarter-wave plate. Others groups favour circular selective scattering (Di Nuzzo et al., *ACS Nano* **11**, 12713 (2017), Sharma et al., *J. Phys. Chem. Lett.* **10**, 7547 (2019)) as another reasonable mechanism for the strong circular polarisation of emission. As already discussed in the reply to reviewer 1, we addressed this point shortly on page 17:

“... As already mentioned, CPL is also thought to be a long-range supramolecular effect, likely resulting from processes such as circular intensity differential scattering^{22,36} and linearly polarised luminescence of quasi-nematic layers with subsequent circular polarisation by the remaining film layer⁴¹. ...”

REVIEWERS' COMMENTS

Reviewer #1 (Remarks to the Author):

The authors very carefully addressed the concerns raised

I recommend publication without further delay

Reviewer #2 (Remarks to the Author):

I thank the authors for thoroughly answering my and the other reviewers' questions. The clarifying changes to the text are very helpful and the additional figures and results clearly support the authors' findings and analysis. Without reserve, I highly recommend publication of this revised paper.

Only a short remark concerning answer 8 to my comments: it is very pleasing to see how the transient CD data imposes additional constraints on the fit, but a kinetic model is of course not proven correct or unique even by the most consistent fitting result.

Reviewer #3 (Remarks to the Author):

I am satisfied with authors' additions and response and recommend publication in the present version.

Response to the reviewers' comments

(Note: Reviewers comments, reproduced verbatim, are in italic font, with central points highlighted in bold. Authors' replies are in normal font.)

Authors' reply to the comment of Reviewer 1

“The authors very carefully addressed the concerns raised I recommend publication without further delay.”

Authors' reply: We are pleased that reviewer 1 is content with our response to the points raised.

Authors' reply to the comment of Reviewer 2

“I thank the authors for thoroughly answering my and the other reviewers' questions. The clarifying changes to the text are very helpful and the additional figures and results clearly support the authors' findings and analysis. Without reserve, I highly recommend publication of this revised paper.

Only a short remark concerning answer 8 to my comments: it is very pleasing to see how the transient CD data imposes additional constraints on the fit, but a kinetic model is of course not proven correct or unique even by the most consistent fitting result.”

Authors' reply: We appreciate that reviewer 2 agrees with all the changes we made to the manuscript. We agree with the reviewer's opinion that a kinetic model is not proven correct or unique simply by showing consistent fitting results. Of course, it is possible to invoke different mechanisms. In our modelling approach, we included those excited-state processes in polymer thin films, which have been proven to occur based on a number of experiments done in other groups and also based on our own experimental observations. In this respect, our approach considers the minimum number of reaction steps in the kinetic scheme (e.g. S_x decay, S_1 decay, bimolecular diffusive recombination, FRET, ...), so “Occam's razor” is employed. Of course, future research on these films might reveal additional kinetic processes currently not considered.

Authors' reply to the comment of Reviewer 3

“I am satisfied with authors' additions and response and recommend publication in the present version.”

Authors' reply: We thank reviewer 3 for the positive comment on our changes.